# Leveraging an ECG Beat Diffusion Model for Morphological Reconstruction from Indirect Signals

**Lisa Bedin**[*†]
Ecole Polytechnique

**Gabriel Cardoso**[†]
Ecole Polytechnique, IHU Liryc

**Josselin Duchateau**
Bordeaux University Hospital, IHU Liryc

**Remi Dubois**
IHU Liryc

**Eric Moulines**
Ecole Polytechnique

## Abstract

Electrocardiogram (ECG) signals provide essential information about the heart's condition and are widely used for diagnosing cardiovascular diseases. The morphology of a single heartbeat over the available leads is a primary biosignal for monitoring cardiac conditions. However, analyzing heartbeat morphology can be challenging due to noise and artifacts, missing leads, and a lack of annotated data. Generative models, such as denoising diffusion generative models (DDMs), have proven successful in generating complex data. We introduce `BeatDiff`, a lightweight DDM tailored for the morphology of multiple leads heartbeats. We then show that many important ECG downstream tasks can be formulated as conditional generation methods in a Bayesian inverse problem framework using `BeatDiff` as priors. We propose `EM-BeatDiff`, an Expectation-Maximization algorithm, to solve this conditional generation tasks without fine-tuning. We illustrate our results with several tasks, such as removal of ECG noise and artifacts (baseline wander, electrode motion), reconstruction of a 12-lead ECG from a single lead (useful for ECG reconstruction of smartwatch experiments), and unsupervised explainable anomaly detection. Experiments show that the combination of `BeatDiff` and `EM-BeatDiff` outperforms SOTA methods for the problems considered in this work.

## 1 Introduction

Electrocardiograms (ECG) are essential tools for diagnosing cardiac conditions. Two main types of diagnostics can be obtained from an ECG signal: rhythm-based and morphology-based. Rhythm-based diagnostics focus on the frequency and regularity of heartbeats, while morphology-based diagnostics focus on the shape and amplitude of the various waves and segments of the ECG signal; see [40]. Many critical cardiac conditions can be diagnosed by analyzing the morphology of the different phases of a single beat ([66, 67, 20, 39]). For example, an increase in the ST segment suggests a myocardial infarction ([93]), while a long QT syndrome is associated with an increased risk of sudden death ([92]). In fact, patients who have survived events similar to sudden death often have abnormal intracardiac signals, even during sinus rhythm; see [34].

Most generative models for ECG literature attempt to accurately represent the rhythm, i.e. the time at which the individual ECG events occur; see e.g., [30, 99, 79, 23, 104, 100, 3]. We focus in this paper on the morphology of a single heartbeat.

---

Code available at `https://github.com/LisaBedin/BeatDiff`

Pretrained models available at `https://huggingface.co/lbedin/BeatDiff`

*Correspondence: `lisa.bedin@polytechnique.org` †Equal contribution.

The standard for ECG recording systems includes 12 leads, which are obtained using 9 electrodes. The use of multiple leads is required because pathologies may manifest only in one lead. For example, T wave inversions—which can indicate Arrhythmogenic Right Ventricula Dysplasia—might only appear on one or two precordial leads ([102]). Furthermore, the specific precordial lead in which a disease is detected has an important diagnostic value about the disease location in the heart ([52]).

Multi-lead ECGs can be affected by noise and artefacts, i.e. unwanted signals caused primarily by variations in potential and impedance at the electrode-skin interface, but also by other factors such as environmental interference, movement of the subject; see e.g., [101, 47, 14, 56] and the references therein. These artefacts overlap in the spectral range of interest and manifest as morphological features that resemble inherent aspects of the ECG or disease-specific aspects. Therefore, methods that can accurately reconstruct 12-lead ECGs from partial observations are central to analysing the morphology of heartbeats. In this work, we address classical problems such as baseline wander or motion artefacts; we are also interested in the reconstruction of missing leads to be able to reconstruct the 12-lead representation from a reduced number of electrodes, which is an essential precursor for the reconstruction of 12-lead ECGs from smartwatch measurements; see [90].

The heartbeat morphology reconstruction problems are naturally formulated as Bayesian linear inverse problems ([87, 38, 21]). The observation vector is an affine transformation of the signal of interest, which is influenced by additive noise. This affine function is only partially known (it may depend on unknown parameters) and the corresponding inverse problem is usually ill-posed (in the missing lead case, the affine function is not invertible). Inverse Bayesian problems require the use of a prior distribution for the signal to be reconstructed. Recently, the use of generative models to define priors has enabled us to achieve many successes in various areas.; see e.g.,[5, 57, 64, 96]. Early works in this area used flow models or GANs. There has been a recent increase in interest in using diffusion models as a prior in Bayesian inverse problems ([83, 17, 85, 45, 46, 13, 98]). Many techniques have been recently proposed. We focus here on `MCGDiff` ([13]), which constructs unbiased estimates of posterior distributions using Sequential Monte Carlo (SMC) methods, a.k.a. particle filter.

Our main contributions are as follow

- We introduce `BeatDiff` a new light-weight DDM model designed to generate 12-leads heart beat morphology. In comparison to [3], `BeatDiff` has a significanty lower memory footprint and faster generation speed. `BeatDiff` has shown superior performance to state-of-the-art ECG generation methods accross all the metrics that we have considered.

- We then show how `BeatDiff` can be used as a prior to address various challenges in heartbeat morphology reconstruction from partial observations. We show how the `MCGDiff` method can be combined with Monte Carlo Expectation Maximization (MCEM) algorithm to compute maximum likelihood estimate of the unknown parameters of the inverse linear model (e.g., noise level estimation, noise and artifact model, etc.), leading to a new full-fledged algorithm for conditional ECG generation, called `EM-BeatDiff`.

- We demonstrate the effectiveness of our approach by comparing it to state-of-the-art methods. Our algorithm outperforms the current best approaches on multiple evaluation metrics specifically designed for ECGs, and offers new paths that have the potential to lead to novel applications.

**Related works:** The use of generative models ([50, 51, 33]) as informative priors in solving Bayesian inverse problems has attracted significant interest ([4, 96, 88, 42, 77, 103, 74]). In particular, DDMs have been demonstrated as a particularly suitable choice of prior for solving inverse problems ([83, 17, 85, 45, 46]). DDMs are generative models that transform a simple reference distribution into the training data distribution through a denoising process called denoising diffusion. These models are capable of generating high-quality realistic samples on par with the best Generative Adversarial Networks (GANs) ([32]) in terms of image and audio generation, without the intricacies of adversarial training ([81, 86, 83, 84, 9]). In this article, we follow the approach proposed in [13, 98], for sampling solutions to an inverse problem using a Sequential Monte Carlo (SMC) algorithm that guides the denoising process of a pretrained diffusion model. This method is accompanied by a series of theoretical guarantees in realistic scenarios.

Generative modeling, denoising methods, and automatic anomaly detection algorithms are commonly used for ECG analysis. In particular, DDMs have been demonstrated to be capable of generating realistic ECGs: [2] focuses on generating a single healthy beat for a single ECG lead, [3] generates a

10-second period conditioned on various complementary ECG information. Additionally, numerous methods address the denoising problem in ECGs; see e.g., [79, 56, 15]. Classical approaches like Dower matrices ([59]) are used to reconstruct missing leads in ECGs. [97, 43] rely on neural networks to detect anomalies, and [76] use adversarial autoencoders for unsupervised anomaly detection. However, to our knowledge, there is no method that addresses all these problems with a single pretrained model.

## 2  `BeatDiff` - a generative model for heartbeat morphology

**Denoising Diffusion Generative Models (DDM):**   We briefly describe in this section the DDMs and introduce some basic notations which are required below; see [24, 37, 86, 83, 85, 44, 19] and the references therein for theory and practical implementation details. We focus on the variance-exploding (VE) framework ([86]). In the *forward path* an initial state $X_0$ is sampled from the data distribution $\mathsf{q}_{\mathrm{data}}$ and independent Gaussian noise with zero-mean and increasing variance is added to generate subsequent states $X_k = X_{k-1} + \rho_k \varepsilon_k$, where $k \in \mathbb{N}^*$, $\rho_k > 0$, and $\varepsilon_k \sim \mathcal{N}(0, \mathrm{I})$. The joint p.d.f. of the Markov chain is $\mathsf{q}_{0:K}(x_{0:K}) = \mathsf{q}_{\mathrm{data}}(x_0) \prod_{k=1}^{K} q_k(x_k|x_{k-1})$, where $q_k(\cdot|x_{k-1}) = \mathcal{N}(x_{k-1}, \rho_k^2 \mathrm{I})$ and $K \in \mathbb{N}^*$. The conditional distribution of $X_k$ given $X_s$ with $k > s \geq 0$ is given by $\mathsf{q}_{k|s}(\cdot|x_s) = \mathcal{N}(x_s, (v_k^2 - v_s^2)\mathrm{I})$ with $v_k^2 = \sum_{j=1}^{k} \rho_j^2$ (and $v_0^2 = 0$). The number of forward steps $K$ is chosen such that $v_K^2 = v_{\max}^2$ is far larger than the variance of $\mathsf{q}_{\mathrm{data}}$. With such choice, $\mathsf{q}_{K|0}(\cdot|x_0)$ is close to the reference distribution $q_{\mathrm{ref}} = \mathcal{N}(0, v_{\max}^2 \mathrm{I})$. We learn for each state $X_k$ a denoiser $\mathcal{D}_{0|k}^{\varphi}$ with parameters $\varphi$ trained to minimize $\mathcal{L}_{\mathcal{D}}(\varphi) := \sum_{k=1}^{K} \gamma_k^2 \mathbb{E}_{X_0 \sim \mathsf{q}_{\mathrm{data}}, \epsilon \sim \mathcal{N}(0,\mathrm{I})} \left[ \|\mathcal{D}_{0|k}^{\varphi}(X_0 + v_k\epsilon, v_k) - X_0\|^2 \right]$, where $\{\gamma_k\}_{k \in [1:K]}$ is a sequence of appropriately defined positive weights. We denote the result of this minimization as $\varphi^*$. In the backward path, we sample $x_K \sim q_{\mathrm{ref}}$ and for $k = K$ to $k = 2$ we sample $x_{k-1}$ given $x_k$ with

$$p_{k-1|k}(x_{k-1}|x_k) = \mathcal{N}\left(x_{k-1}; \boldsymbol{\mu}_{k-1}(x_k, \mathcal{D}_{0|k}^{\varphi^*}(x_k, v_k)), \eta_{k-1}^2 \mathrm{I}_d\right)$$

where the variances are hyperparameters $\eta = \{\eta_k\}_{k \in \mathbb{N}}$ satisfying $\eta_k^2 \leq v_k^2$ and $\boldsymbol{\mu}_{k-1}(x_k, x_0) := x_0 + (v_{k-1}^2/v_k^2 - \eta_{k-1}^2/v_k^2)^{1/2}(x_k - x_0)$. Finally, we sample $x_0 \sim p_0(\cdot|x_1) := \mathcal{N}(\mathcal{D}_{0|1}^{\varphi^*}(x_1, v_1), \eta_0^2 \mathrm{I})$. To keep the notations simple, we remove in the sequel the dependence in $\eta$ and $\varphi^*$. For $k \in [0 : K-1]$, we denote by $\mathsf{p}_k(x_k)$ the marginal distribution of $X_k$:

$$\mathsf{p}_k(x_k) := \int q_{\mathrm{ref}}(x_K) \prod_{s=K}^{k+1} p_{s-1|s}(x_{s-1}|x_s) \mathrm{d}x_{k+1:K}.$$

`BeatDiff` **model:**   In the standard ECG, the augmented limb leads (AVL, AVR, AVF) can be obtained from a known linear combination of the limb leads (I, II, III) ([59, Vol 1, Chapter 11]). Hence, it is standard practice to select either the augmented leads or the limb leads to model the ECG ([3, 35]). We exclude the augmented leads and use the leads (I, II, III, V1–6). We denote by $L = 9$ the number of leads, and by $T$ the maximal heartbeat duration (expressed in number of samples).

Various factors, including age (A), sex (S) and the RR interval, which is the reciprocal of heart rate, influence the morphology of the heartbeat; see e.g., [60, 75, 6]. Therefore, we use the DDM described above to approximate the distribution $\mathsf{q}_{\mathrm{data}}$ of heartbeats over the retained leads *conditionally* on the patient characteristics $\mathcal{P} := (A, S, RR)$. The denoiser of `BeatDiff` $\mathcal{D}_{0|k}^{\varphi}$ takes as input $(x, v_k, e_{\mathcal{P}})$ where $x$ is the $L \times T$ matrix of single heartbeat samples, $v_k$ is the $k$-th step diffusion variance and $e_{\mathcal{P}}$ encodes the patient features. We obtain $e_{\mathcal{P}}$ from $\mathcal{P} = (A, S, RR)$ by first one-hot-coding the Boolean variable S and then embedding it using a fully connected 2-layer network. For $v_k$ we use the Fourier positional encoding ([91]) of $\log(v_k)$ as in [24]. For $\mathcal{D}_{0|k}^{\varphi}$ we use a modified 1d Unet, the specific details are given in Appendix B.1.4. The model has $10^6$ parameters. Compared to [3], the inference time is 400 times shorter and the memory footprint is 900 times smaller.

`BeatDiff` **training**   We utilize the PhysioNet Challenge dataset ([31, 68, 69]), comprising 43,101 12-lead ECGs. The pre-processing of [8] is used which consists of normalization of the sampling frequency, detection of R peaks to identify heartbeats, segmentation of the heartbeats. We obtain 214,460 single-beat ECGs, each with $T = 176$ samples and $L = 9$ (I, II, III, V1–V6). See Appendix B.1.1 for details.   Each patient (and *the entirety of its recordings*) is attributed to one of the three datasets: Training, Cross-validation (CV) or Test. During training, a batch of size $b$

is constituted by firstly drawing $b$ patients and then selecting randomly one of the beats for each given patient. For testing and cross-validation, due to the significant variability between patients in comparison to the variability between heartbeats, we randomly select a single beat per patient for model evaluation. The entire network $\mathcal{D}^{\varphi}_{0|k}$ is trained to minimize $\mathcal{L}_{\mathcal{D}}$ using the Adam optimizer ([49]) with a batch-size $2^{10}$ on the healthy training set, and the best model in terms of $\mathcal{L}_{\mathcal{D}}$ over the cross-validation set is retained. See Appendix B.1.4 for details.

## 3  EM-BeatDiff - conditional heartbeat generation from indirect measurements

We present EM-BeatDiff a method that allows us to sample heartbeat morphology from partial observations, focusing on a class of problems that can be formulated as Bayesian linear inverse problems. Our approach is based on Monte-Carlo guided diffusion (MCGDiff), introduced in [13] (see also [89, 65]), which is used in combination with BeatDiff.

**Monte Carlo Guided Diffusion (MCGDiff):**   In many applications of interest, the objective is to sample from a distribution $\phi_0(x_0) := g_0(x_0)\mathsf{p}_0(x_0)/\mathcal{Z}$, where $g_0$ is a nonnegative potential function, $\mathsf{p}_0$, the marginal of the diffusion model at time 0, and $\mathcal{Z} := \int g_0(x)\mathsf{p}_0(x)\mathrm{d}x$ is the normalizing constant. For example, in a Bayesian setting, $g_0$ is the likelihood function (the conditional distribution of the observation given the current value of the state $x_0$) and $\mathsf{p}_0(x_0)$ is the prior distribution of the state. In such case, $\phi_0(x_0)$ is the posterior distribution of the state $x_0$ given the current observation. A simple idea for sampling the posterior $\phi_0(x_0)$ is to use sampling importance resampling (SIR, [72]), where $\mathsf{p}_0$ is used as the instrumental distribution. However, this method may be inefficient since the instrumental distribution neglects the potential $g_0$.

We define a distribution over the path space

$$\phi_{0:K}(x_{0:K}) := \mathcal{Z}^{-1} g_0(x_0) \prod_{k=1}^{K} p_{k-1|k}(x_{k-1}|x_k)q_{\mathrm{ref}}(x_K).$$

In [13], a sequence of positive intermediate potentials $\{g_k\}_{k\in[1:K]}$ with $g_K \equiv 1$ was introduced to guide the backward Markov chain to regions where the potential $g_0$ is large. The path space distributions may be equivalently rewritten as

$$\phi_{0:K}(x_{0:K}) \propto q_{\mathrm{ref}}(x_K) \prod_{k=1}^{K} \frac{g_{k-1}(x_{k-1})}{g_k(x_k)} p_{k-1|k}(x_{k-1}|x_k)$$
$$\propto q_{\mathrm{ref}}(x_K) \prod_{k=1}^{K} \omega_k(x_k)\hat{p}_{k-1|k}(x_{k-1}|x_k)\,, \tag{3.1}$$

where, for $k \in [1 : K]$, $\hat{p}_{k-1|k}(\cdot|x_k) := g_{k-1}(\cdot)p_{k-1|k}(\cdot|x_k)/\mathcal{Z}_k(x_k)$, and $\mathcal{Z}_k(x_k) := \int g_{k-1}(x')p_{k-1|k}(x'|x_k)\mathrm{d}x$, and $\omega_k(x_k) := \mathcal{Z}_k(x_k)/g_k(x_k)$. We implicitly assume that these formulas have a closed form. Sampling according to (3.1) passes through the intermediate distributions $\phi_k(x_k) := \int q_{\mathrm{ref}}(x_K) \prod_{s=k+1}^{K} \omega_s(x_s)\hat{p}_{s-1|s}(x_{s-1}|x_s)\mathrm{d}x_{k+1:K}$ for each $k \in [1 : K]$, which verifies

$$\phi_{k-1}(x_{k-1}) \propto g_{k-1}(x_{k-1})\mathsf{p}_{k-1}(x_{k-1}) \propto \int \omega_k(x)\hat{p}_{k-1|k}(x_{k-1}|x)\phi_k(x)\mathrm{d}x\,. \tag{3.2}$$

Each $\phi_{k-1}$ thus has the same structure as $\phi_0$: a product of a potential function and the marginal law at time $k-1$ of the backward diffusion.

It remains to approximate this sequence of distributions. For this purpose, we use Sequential Monte Carlo (SMC); see [25, 16]. Suppose that we have at iteration $k$ a *particle approximation* $\phi_k^M = M^{-1} \sum_{j=1}^{M} \delta_{\xi_k^j}$ of $\phi_k$ through a set of $M \in \mathbb{N}_{>0}$ particles $\xi_k^{1:M}$, initialized with $\xi_K^{1:M} \sim q_{\mathrm{ref}}^{\times M}$. Plugging this approximation into Equation (3.2) gives

$$\phi_{k-1} \propto \sum_{j=1}^{M} \omega_k(\xi_k^j)\hat{p}_{k-1|k}(\cdot|\xi_k^j)\,.$$

Hence, to obtain $\xi_{k-1}^{1:M}$, we first sample $M$ ancestors according to $I_{k-1}^{1:M} \sim \mathrm{Cat}\big(\{\omega_k(\xi_k^j)/\sum_{i=1}^{M}\omega_k(\xi_k^i)\}_{j=1}^{M}\big)^{\times M}$, then we sample new particles $\xi_{k-1}^{1:M} \sim \{\hat{p}_{k-1|k}(\cdot|\xi_k^{I_{k-1}^j})\}_{j=1}^{M}$, leading to $\phi_{k-1}^M = M^{-1} \sum_{j=1}^{M} \delta_{\xi_{k-1}^j}$. Algorithm is given in Appendix A.1.1.

**Bayesian inverse problems** We assume that the $d_y \times 1$ vector of observations $Y$ (noisy/partial heartbeat) is given by

$$Y = A_\theta X_0 + b_\theta + D_\theta \epsilon, \quad \epsilon \sim \mathcal{N}(0, I), \quad X_0 \sim p_0 \tag{3.3}$$

where $A_\theta$ is a $d_y \times d_x$ matrix (for selecting the lead/time observed indices), $b_\theta$ is a $d_y \times 1$ vector (modeling hearbeat artifacts; e.g., baseline wander, electrode motion), $D_\theta$ is a $d_y \times d_y$ invertible matrix (the variance of the noise), and $\theta \in \Theta$ is a vector of unknown parameters. Define by $g_0^{\theta,y}(x_0)$ the likelihood of the observation, given by $g_0^{\theta,y}(x_0) := \mathcal{N}(y; A_\theta x + b_\theta, D_\theta D_\theta^\top)$. Given an observation $y$ and a value of the parameter $\theta$, we may sample $X_0$ from the posterior $X_0|y,\theta$, with p.d.f. $\phi_0^{\theta,y}(x_0) = p_0(x_0) g_0^{\theta,y}(x_0)/\mathcal{Z}^{\theta,y}$, $\mathcal{Z}^{\theta,y} = \int g_0^{\theta,y}(x) p_0(x) \mathrm{d}x$ is the normalizing constant. We use MCGDiff with the intermediate potentials $\{g_k^{\theta,y}\}_{k \in [0:K]}$ defined as

$$g_k^{\theta,y}(x) = \mathcal{N}(y; A_\theta x + b_\theta, \Sigma_{k,\theta}), \tag{3.4}$$

where the sequence of covariance matrices $\Sigma_{k,\theta}$ are specified in Appendix A.2.1. For this choice of potentials, $\hat{p}_{k-1|k}^{\theta,y}$ and $\omega_k^{\theta,y}$ admit closed forms given in Appendix A.2.2.

MCGDiff allows to sample $X_0|y,\theta$ for a known parameter $\theta$. When $\theta$ is unknown, we maximize the penalized marginal log-likelihood

$$\theta^* = \underset{\theta \in \mathbb{R}^{\bar{d}}}{\arg\max} \left( l(\theta) + \mathrm{Pen}(\theta) \right), \quad l(\theta) := \log \mathcal{Z}^{\theta,y} = \log \int g_0^{y,\theta}(x) p_0(x) \mathrm{d}x \tag{3.5}$$

where $\mathrm{Pen}(\theta)$ is a penalty. The best-known method for optimizing the marginal log-likelihood (3.5) is the expectation maximization algorithm (EM); see [61]. The EM iterates between two main steps: expectation (E) and maximization (M). Starting from an initial guess $\theta_0$, the EM algorithm alternates between: (E) compute the surrogate function $Q(\theta; \theta_i) := \int \log g_0^{y,\theta}(x_0) \phi_0^{\theta_i,y}(x_0) \mathrm{d}x_0$; and (M) solve for $\theta_{i+1} := \arg\max_{\theta \in \Theta} Q(\theta; \theta_i) + \mathrm{Pen}(\theta)$. Under general conditions, the sequence of parameter estimates $(\theta_i)_{i \in \mathbb{N}}$ converges to a stationary point $\theta_*$ of the marginal penalized likelihood; see [61, Chapter 3]. In this setting, the E-step is untractable; we approximate the surrogate function in the (E) step using MCGDiff with the current parameter $\theta_i$ and the sequence of intermediate potentials defined in (3.4). Such scheme becomes a specific instance of the Monte Carlo EM algorithm (MCEM), initially introduced in [95] and further analyzed in [54, 26, 53].

**The** EM-BeatDiff **algorithm:** We combine the BeatDiff for the prior and MCGDiff algorithms for posterior sampling, with MCEM steps for parameter inference. The only slight difference is that the observations are gathered in a matrix $\mathbf{Y}$ of size $\tilde{L} \times \tilde{T}$ - where $\tilde{L}$ is the number of observed leads and $\tilde{T}$ the number of observed samples on each lead. The state we are attempting to reconstruct is a matrix of size $L \times T$. The observation equation takes the form

$$Y = A_\theta X_0 \bar{A}_\theta + B_\theta + D_\theta \epsilon \bar{D}_\theta,$$

where $(A_\theta, D_\theta)$ and $(\bar{A}_\theta, \bar{D}_\theta)$ are $\tilde{L} \times L$ and $T \times \tilde{T}$ matrices, $B_\theta$ is a $\tilde{L} \times \tilde{T}$ matrix, and $\epsilon$ is a $L \times T$ matrix with i.i.d. standard Gaussian entries. The model (3.3) is obtained by applying the vectorization operator to $\mathbf{Y}$. The full algorithm is detailed in Appendix A.3. Note that EM-BeatDiff is also applicable to "standard" ECG signals and could be used in combination with the ECG generative model of [3], for example.

## 4 Experimental validation

BeatDiff **evaluation:** We begin by assessing the impact of BeatDiff in a classifier improvement task. This involves comparing the performance of a classifier trained on a severely unbalanced dataset with that of the same classifier trained on a balanced dataset. The balancing is achieved by augmenting the minority class with new examples from a generative model. Results for sex classification from heartbeat are reported in Table 1, including model size, inference time, F1 score, total accuracy, and AUC score are shown for the balanced dataset with BeatDiff and alternative ECG generation models from [1, 3], retrained to generate ECGs conditioned on sex. Diffusion-based models BeatDiff and SSDM ([3]) outperform WGAN ([1]), with BeatDiff being 400 times faster than [3]. See implementation details in Appendices B.3.1, B.3.2 and B.4.

Table 1: Evaluation of ECG generation models for balancing sex-imbalanced datasets in heartbeat classification task. F and M refer to the number of female and male real heartbeats in the training set. Confidence intervals are obtained by re-initializing the classifier training and the generated data used to balance the datasets.

| Model | Size (Mb, ↓) | Inference Time (ms, ↓) | F = 10%M F1 (%, ↑) | F = 10%M Acc. (%, ↑) | F = 10%M AUC (%, ↑) | F = 5%M F1 (%, ↑) | F = 5%M Acc. (%, ↑) | F = 5%M AUC (%, ↑) |
|---|---|---|---|---|---|---|---|---|
| SSDM [3] | $39 \times 10^3$ | $7.5 \times 10^4$ | $76 \pm 0$ | $69 \pm 1$ | $77 \pm 1$ | $74 \pm 0$ | $64 \pm 0$ | $71 \pm 1$ |
| WGAN [1] | 27 | $3.8 \times 10^{-2}$ | $76 \pm 0$ | $69 \pm 1$ | $77 \pm 1$ | $74 \pm 0$ | $64 \pm 1$ | $72 \pm 1$ |
| BeatDiff | 42 | $1.6 \times 10^2$ | $\mathbf{78 \pm 0}$ | $\mathbf{73 \pm 1}$ | $\mathbf{79 \pm 1}$ | $\mathbf{77 \pm 1}$ | $\mathbf{72 \pm 1}$ | $\mathbf{76 \pm 1}$ |
| Unbalanced | - | - | $76 \pm 0$ | $69 \pm 1$ | $74 \pm 3$ | $74 \pm 1$ | $64 \pm 1$ | $70 \pm 3$ |
| Balanced | - | - | $82 \pm 0$ | $81 \pm 0$ | $86 \pm 1$ | $82 \pm 0$ | $81 \pm 0$ | $86 \pm 1$ |

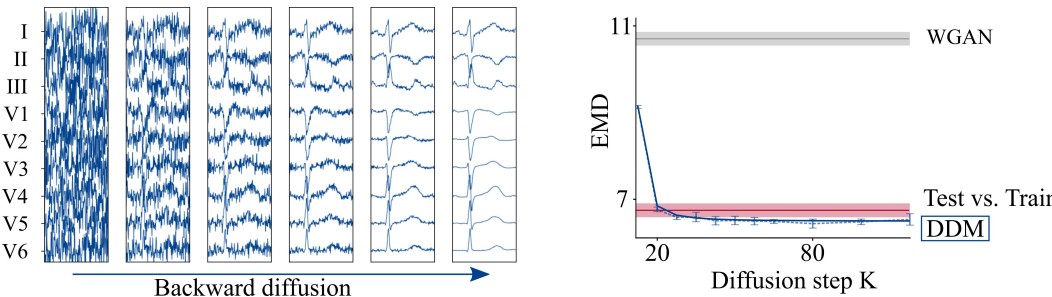

Figure 1: *Left:* heartbeat generation along backward diffusion steps. *Right:* EMD between generated ECG distribution and real ECG distribution. EMD vs. test (resp. train) in plain (resp. dotted) line. EMD for DDM with different number of diffusion steps, in blue. DDM for WGAN model in gray. EMD between test and train distributions in red. Error bars correspond to different training batches of size 2864.

We also use the L2-Earth Mover's Distance (EMD) ([29]) to evaluate the dissimilarity between the predicted and target distributions, excluding SSDM due to computational limitations. The EMD is computed from the generated set for both the test set and batches of the training set of the same size. Our results in figure 1 show that a few diffusion steps are sufficient to generate an accurate prediction distribution, with BeatDiff performing better than [1] in replicating the real data distribution. In Appendix B.6.1 we present a third evaluation of the generated ECGs' quality using the out-of-distribution score proposed in [18].

**Prediction of Corrected QT:** Both the EMD and Classifier Enhancement tasks are concerned with how different the generated ECGs are from the ECGs in the dataset. We are now focusing on the question "Is the algorithm able to correctly capture underlying physiological mechanism?". To do so, we evaluate EM-BeatDiff on the prediction of corrected QT, which is an important clinical indicator obtained from the ECG; see [7].

The QT interval is the duration between the Q wave, which marks the beginning of ventricular depolarization, and the end of the T wave, which signifies the completion of ventricular repolarization. This interval depends on heart rate: as the heart rate increases (and the RR interval decreases), the QT interval tends to shorten. Understanding the relationship between the RR and QT intervals is crucial for diagnosing and managing various cardiac conditions. For instance, a prolonged QT interval may indicate an increased risk of arrhythmias, such as Torsades de Pointes, while a shortened QT interval can be associated with conditions like hypercalcemia. Moreover, certain medications are known to prolong the QT interval; see [58].

We use EM-BeatDiff to generate the T-Wave from a given patient QRS complex (the sequence of waves (Q, R, S) with negative, positive, and negative deflections, respectively) and heart rate (RR). Each test ECG is trimmed to focus solely on the QRS complex. Then, for RR values ranging from 0.6 s to 1.2 s, or equivalently for heart rates ranging from 43 to 100 beats per minute, we sample $x$ from the conditional distribution of the ECGs over all leads given the RR and the observed QRS as illustrated in figure 2. The configuration of the related inverse problem is given in Table 2 under the name QT.

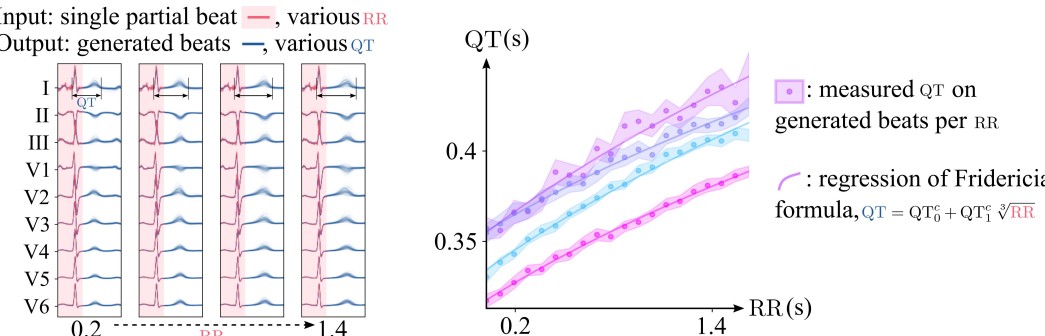

Figure 2: *Left:* Example of T-wave prediction (blue) conditioned on Q-wave (red) for different value of RR. *Right:* QT as a function of RR for 4 patients. QT measured in 100 generated samples (resp. regressed with Fridericia formula) displayed in dots with 95%-CLT bars (resp. curve).

Table 2: Configurations used for `EM-BeatDiff` for each task.

| Task | $(\bar{L}, \bar{T})$ | $\theta$ | $A_\theta$ | $\bar{A}_\theta$ | $B_\theta$ | $D_\theta$ | $\bar{D}_\theta$ |
|---|---|---|---|---|---|---|---|
| QT | $(L, 70)$ | $\sigma_{1:L}$ | $\mathrm{I}_{L \times L}$ | $\mathrm{I}_{T \times \bar{T}}$ | $\mathbf{0}_{L,\bar{T}}$ | $\mathrm{diag}(\sigma_{1:L})$ | $\mathrm{I}_{T \times \bar{T}}$ |
| AR | $(L, T)$ | $(\sigma_{1:L}, \vartheta_{1:K,1:L})$ | $\mathrm{I}_{L \times L}$ | $\mathrm{I}_{T \times T}$ | (4.1) | $\mathrm{diag}(\sigma_{1:L})$ | $\mathrm{I}_{T \times T}$ |
| ML (SW) | $(1, T)$ | $\sigma$ | $\mathrm{I}_{\bar{L} \times L}$ | $\mathrm{I}_{T \times T}$ | $\mathbf{0}_{\bar{L},T}$ | $\sigma \, \mathrm{I}_{\bar{L} \times L}$ | $\mathrm{I}_{T \times T}$ |
| ML (V1-6) | $(3, T)$ | $\sigma_{1:\bar{L}}$ | $\mathrm{I}_{\bar{L} \times L}$ | $\mathrm{I}_{T \times T}$ | $\mathbf{0}_{\bar{L},T}$ | $\mathrm{diag}(\sigma_{1:\bar{L}}) \, \mathrm{I}_{\bar{L} \times L}$ | $\mathrm{I}_{T \times T}$ |
| AD (MI) | $(3, T)$ | $(\sigma_{1:\bar{L}}, \vartheta_{1:K,1:\bar{L}})$ | $\mathrm{I}_{\bar{L} \times L}$ | $\mathrm{I}_{T \times T}$ | (4.1) | $\mathrm{diag}(\sigma_{1:\bar{L}}) \, \mathrm{I}_{\bar{L} \times L}$ | $\mathrm{I}_{T \times T}$ |
| AD (LAE) | $(3, T)$ | $(\sigma_{1:\bar{L}}, \vartheta_{1:K,1:\bar{L}})$ | $\mathrm{I}_{\bar{L} \times L}$ | $\mathrm{I}_{T \times T}$ | (4.1) | $\mathrm{diag}(\sigma_{1:L}) \, \mathrm{I}_{\bar{L} \times L}$ | $\mathrm{I}_{T \times T}$ |
| AD (LAD) | $(L, 106)$ | $\sigma_{1:L}$ | $\mathrm{I}_{L \times L}$ | $\left[ \mathbf{0}_{\bar{T},T-\bar{T}} ; \mathrm{I}_{\bar{T} \times \bar{T}} \right]^T$ | (4.1) | $\mathrm{diag}(\sigma_{1:L}) \, \mathrm{I}_{\bar{L} \times L}$ | $\mathrm{I}_{T \times \bar{T}}$ |
| AD (LQT) | $(L, 70)$ | $\sigma_{1:L}$ | $\mathrm{I}_{L \times L}$ | $\mathrm{I}_{T \times T}$ | $\mathbf{0}_{L,\bar{T}}$ | $\mathrm{diag}(\sigma_{1:L}) \, \mathrm{I}_{L \times L}$ | $\mathrm{I}_{T \times \bar{T}}$ |

To evaluate this, we rely on well-known empirical formulas from [7, 27, 73]. These formulas introduce coefficients called "corrected QT" denoted as $\mathrm{QT}_0^c$ and $\mathrm{QT}_1^c$, which depend on the patient and are determined from ECGs measured during stress test. We regress the intercept $\mathrm{QT}_0^c$ and slope $\mathrm{QT}_1^c$ of the Fridericia formula from [27], which states that $\mathrm{QT} = \mathrm{QT}_0^c + \mathrm{QT}_1^c \sqrt[3]{\mathrm{RR}}$, from the generated curves. As shown in figure 2, we observe a consistent trend between the observed and regressed curves for four patients. Additionally, Table 10 indicates a high $R^2$-score of 0.98 between observed and expected QT curves in the test set. `EM-BeatDiff` generates QT for different RR that follow the Fridericia formula, one of the most correlated with patient QT vs. RR behaviour, without explicitly encoding this relationship during training. This demonstrates the ability of `EM-BeatDiff` to capture underlying physiological mechanisms.

**Artifact removal (AR):** Many solutions for removing ECG artifacts such as baseline wander–a low-frequency artifact caused mainly by respiration and body movements– or electrode motion–also a low-frequency artifact caused by bad electrode contact– have been proposed so far, most often based on adaptive filter, time-frequency (and most notably empirical mode decomposition) and time-scale decomposition; see [22, 94, 101, 55, 14] and the references therein. We use in this experiment a sparse representation of the artifacts in a dictionary, and propose to use penalized MLE with DDM prior to estimate the artifacts and denoise the ECG. In this case, we set for a given $\bar{L} \in \{1, L\}$ and $\bar{T} \in \{1, T\}$

$$B_\theta = [b_1^\theta, \ldots, b_{\bar{L}}^\theta]^\top \quad \text{with} \quad b_{\ell,t}^\theta = \textstyle\sum_{i=1}^K \vartheta_{j,\ell} c_j(t) \quad \text{for} \quad \ell, t \in [1 : \bar{L}] \times [1 : \bar{T}], \qquad (4.1)$$

with $\{c_j\}_{j=1}^J$ be a known set of functions (such as B-splines, a Fourier or a wavelet basis). We choose a Fourier basis in the experiments as expressed in Equation (B.1). The other parameters are given in Table 2 under the name AR.

We therefore remove the artifacts by subtracting a vector assumed to have a sparse representation on an appropriate basis. We use the sparse group LASSO penalty defined as

$$\mathrm{Pen}(\theta) = \lambda_1 \textstyle\sum_{j=1}^J \left( \sum_{\ell=1}^L (\vartheta_{j,\ell})^2 \right)^{1/2} + \lambda_2 \sum_{j=1}^J \sum_{\ell=1}^L |\vartheta_{j,\ell}| \qquad (4.2)$$

which leads to parsimony at both the group and individual levels, in order to promote the selection of the same functions (e.g. Fourier frequency) over all the leads; see [28, 78]. The first term promotes

Table 3: Evaluation of several reconstruction metrics for the AR task on beats corrupted with artifacts from MIT-BIH database from [62], with 95%-CLT intervals over the test-set.

| Method | Baseline Wander | | | Electrode Motion | | |
|---|---|---|---|---|---|---|
| | SSD ($\downarrow$) | MAD ($\downarrow$) | Cos. ($\times 100$, $\uparrow$) | SSD ($\downarrow$) | MAD ($\downarrow$) | Cos. ($\times 100$, $\uparrow$) |
| DeScoD [3] | $4.37 \pm 8.19$ | $0.31 \pm 0.15$ | $95.20 \pm 0.36$ | $0.27 \pm 0.01$ | $0.27 \pm 0.01$ | $92.73 \pm 0.25$ |
| EM-BeatDiff | $\mathbf{0.14 \pm 0.01}$ | $\mathbf{0.24 \pm 0.02}$ | $\mathbf{96.69 \pm 0.22}$ | $\mathbf{0.18 \pm 0.01}$ | $\mathbf{0.26 \pm 0.01}$ | $\mathbf{95.42 \pm 0.19}$ |

Table 4: Evaluation of ECG generation models for the missing lead retrieval task, with 95%-CLT intervals over the test-set.

| Method | Smartwatch | | | V1–6 | | |
|---|---|---|---|---|---|---|
| | SSD ($\downarrow$) | MAD ($\downarrow$) | Cos. ($\times 100$, $\uparrow$) | SSD ($\downarrow$) | MAD ($\downarrow$) | Cos. ($\times 100$, $\uparrow$) |
| EkGAN [41] | $1.63 \pm 0.47$ | $0.36 \pm 0.03$ | $\mathbf{91.33 \pm 0.38}$ | $2.10 \pm 0.83$ | $\mathbf{0.35 \pm 0.03}$ | $\mathbf{93.42 \pm 0.00}$ |
| EM-BeatDiff | $\mathbf{1.03 \pm 0.05}$ | $\mathbf{0.35 \pm 0.01}$ | $86.02 \pm 0.99$ | $\mathbf{1.10 \pm 0.06}$ | $0.36 \pm 0.01$ | $87.78 \pm 0.98$ |

group sparsity: it keeps or removes the projections of observations on $c_j$ across all leads. The second term promotes global sparsity. Details are given in Appendix B.2.2.

The evaluation of EM-BeatDiff in the AR task on 12-lead ECGs contaminated with per-lead independent noise from the MIT-BIH Noise Stress Test database ([62]) is presented in Table 3. Despite not being exposed to MIT-BIH Noise during training, EM-BeatDiff outperforms DeScoD ([56]) –a conditional DDM specifically trained to remove baseline wander– according to the following metrics: Sum of the square of the distances (SSD), Absolute maximum distance (MAD), and Cosine similarity (Cos.) ([63], Appendix B.5). Visualizations of ECGs obtained with EM-BeatDiff and DeScoD are provided in figure 3, and failure cases of DeScoD are discussed in Appendix B.6.4.

**Missing Leads Reconstruction (ML):** In resource-limited clinical settings like ambulatory care, electrode placements can vary from six-lead montages and reduced Frank or EASI configurations to single-lead setups. Similarly, in non-clinical settings, smartwatches like the Apple Watch provide a single lead ECG by measuring the potential between the wrist and the finger of the opposite hand. Recent studies such as [90] showed that this single lead ECG is essentially equivalent to the lead I ECG recorded in a 12 lead ECG. Several papers have addressed the reconstruction of ECGs from a single lead with deep learning, indicating potential applicability in ECG reconstruction from smartwatch single-lead ECG; see e.g., [82, 80, 41].

In the first experiment, we evaluate the performance of EM-BeatDiff in reconstructing V1-6 from the limb leads (I, II, III). This corresponds to the setting ML (V1-6) in Table 2. The second task we consider is generating II, III, V1-6 from lead I, which we refer to as the Smartwatch task. This corresponds to setting ML (SW) in Table 2. In Table 4, EM-BeatDiff outperforms EkGAN ([41]), a deep learning-based methods designed and trained for ECG missing lead reconstruction according to SSD, MAD and Cos. Unlike EkGAN, EM-BeatDiff does not require any task-specific training.

**Cardiac Anomaly Detection (AD):** In this section, we propose using EM-BeatDiff for detecting cardiac abnormalities. Our evaluation methodology consists of evaluating EM-BeatDiff 's capacity to detect four distinct medical conditions: Myocardial Infarction (MI), Left Anterior Descending artery (LAD), Left Atrial Enlargement (LAE), and Long QT syndrome (LQT). These anomalies were selected due to their typical association with localized alterations in P-Wave, QRS, or T-Wave morphologies. To incorporate patient-specific ECG data, we consider three distinct conditioning settings: (I, II, III), QRS, and ST. Conditioning on the limb leads (I, II, III) suggests that the abnormality is more prominently manifested in the precordial leads than the limb leads. Conditioning on QRS indicates that the abnormality is evident in the T-wave, while conditioning on the ST segment implies that the abnormality is present either in the QRS or the P-wave.

For each conditioning type and medical condition, we generate samples from the posterior distribution using EM-BeatDiff. Th anomaly score is the $1 - R^2$ metric between the mean of the generated ECGs from the posterior distribution and the observed ECG over the non-conditional ECG segment.

First, we conduct an ablation study to determine the optimal setting for each medical condition, as detailed in Appendix B.6.3. The chosen optimal settings are presented in Table 2, designated as AD

Table 5: AUC obtained using the proposed anomaly detection score $(1 - R^2)$ for each medical conditioning. See Table 2 for details on the inverse problem in hand. Confidence intervals are obtained by running 10 times `EM-BeatDiff` per heartbeat.

| Model | MI | LAD | LAE | LQT |
|---|---|---|---|---|
| AAE [76] | 80.23 | 82.69 | 74.87 | 70.96 |
| `EM-BeatDiff` | $84.82 \pm 0.01$ | $93.06 \pm 0.03$ | $79.02 \pm 0.07$ | $84.73 \pm 0.04$ |

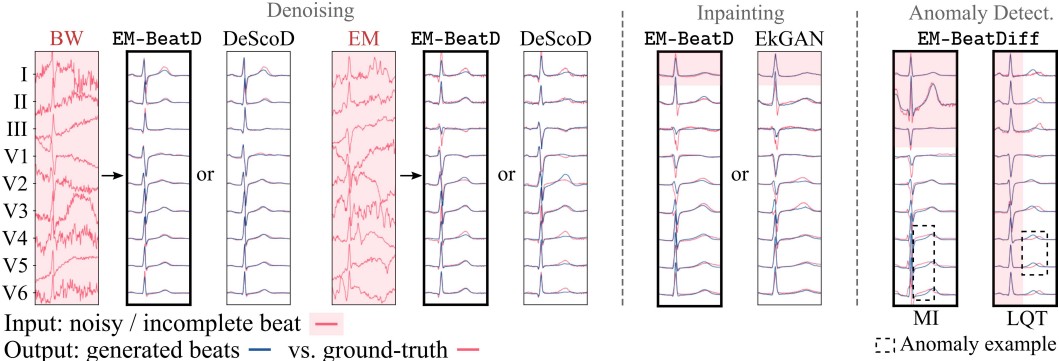

Figure 3: Illustration of `EM-BeatDiff` on the denoising, inpaiting and anomaly detection tasks. The red background indicate the parts of the ECG that are observed through $y$. The red ECGs corresponds to the real ECG and the blue ECGs corresponds to each algorithm reconstructed ECG.

followed by the respective medical condition name. Table 5 shows that `EM-BeatDiff` outperforms AAE ([76]), which uses as anomaly score the MSE between the output of an Adversarial Auto Encoder and the input ECG, according to the AUC of the anomaly score. See Appendix B.3.5 for implementation details. A key advantage of the aforementioned approach is its ability to function in an interactive manner, unlike methods that rely on training on a specific setting. By simply selecting a different set of ECG leads for conditioning, the posterior can be regenerated in a near-online fashion following visual assessment of the posterior and the patient ECGs as in the MI case in Figure 3.

## 5    Conclusion

In this work, we have described a flexible method that addresses several challenges in heart beat morphology, including baseline wander and electrode-motion removal, missing lead reconstruction and anomaly detection, all formulated as Bayesian linear inverse problems. Our method utilises `BeatDiff` a DDM pre-trained to generate the heartbeat morphology on 12 leads, as a prior for sampling solutions to inverse problems with `EM-BeatDiff`. Several evaluation metrics show the effectiveness of `EM-BeatDiff` compared to baseline solutions, which contrary to `EM-BeatDiff` require specific training for the specific task in hand.

Our approach also enables new applications, such as generating a 12-lead ECG using a subset of electrodes, including 12-lead heartbeat morphology reconstruction from smartwatch measurements as an example. Another example is the anomaly detection algorithm which can enable diagnostic of long QT syndrome or other diseases that specifically alter repolarization. In this paper, `BeatDiff` was trained only on healthy ECGs. However, `BeatDiff` could be trained on a dataset containing ECGs presenting pathologies by conditioning on the specific pathology as discussed below.

## 6    Discussion

This paper focus on generating 12-lead healthy heartbeats from partial measurements, e.g., limb leads only, samples corrupted with eletrocde artifacts, see Tables 3 and 4. We show that `EM-BeatDiff` can also be used to classify abnormal heartbeats in an unsupervised manner: we generate healthy counterparts of abnormal heartbeats and use the distance as an anomaly score. The flexibility of our

method allows the detection of various heart conditions (MI, LQT, LAD, LAE) by reconstructing 12 leads from limb leads only, QRS only, or ST only, see Tables 11 and 12.

**Risk of hallucination**    We would like to point out that this anomaly detection tool is semi-white-box: rather than outputting an abnormality score, our approach is also able to show the healthy counterparts of an abnormal signal and highlight where they differ from the patient's signals. This could theoretically enable cardiologists to rule out abnormalities that are not relevant. However, there is still a risk of hallucinations. We have shown that the generated signals are close to the real signals for healthy patients, but a clinical study must be performed before clinical use.

**10s ECG signals**    Although our study focused on heartbeat morphology, we would like to discuss the feasibility of generating realistic longer samples that are more than a beat. We trained a diffusion model to produce 5-second ECGs. We then applied our sampling algorithm to reconstruct 12-lead ECGs from limb leads only and limb leads + V2 + V4. This second setting is similar to AliveCor's recent system Kardia12L*. We found that reconstructing 12 leads from multi-beat ECGs is more complex and requires precordial leads for reasonable results (see figure 11). A special study is needed to build a relevant diffusion model and adapt the algorithm's parameters in order to generate longer 12-lead ECGs from limb leads only. This adaptation is complex due to the need for larger models and more particles for posterior sampling.

**Arrhythmic data**    Testing `EM-BeatDiff`'s ability to reconstruct arrhythmic ECGs is valuable as it would enable the use of additional simulated leads for abnormality detection from portable devices such as smartwatches or AliveCor products. We trained a diffusion model on 5-second ECGs with Atrial Fibrillation (AF) from PhysioNet and successfully predicted a reasonable 12-lead ECG from limbs + V2 + V4 (Kardia12L setting) (see figure 12). A larger AF dataset would yield more interesting results, but this experiment shows `EM-BeatDiff`'s potential for generating rhythmic abnormalities.

**Heartbeat segmentation**    We would like to discuss the use of external segmentation tools for segmenting heartbeats before applying `EM-BeatDiff`. The segmentation of heartbeats is a common practice in the literature. Many works, including the baselines we analyzed in our paper, such as [56], and more recent papers [48], rely on external tools to segment the common 10-second clinical ECG signal into heartbeats. The heartbeat settings of `EM-BeatDiff` could also be used for arrhythmic data such as Premature Ventricular Contractions (PVC), even if they significantly alter the ECG phase. Indeed, one could detect PVCs using available methods such as [12] and remove them from inference.

# 7    Broader Impact

We demonstrate various ways in which `BeatDiff` and `EM-BeatDiff` can be utilized to address various heartbeat morphology analysis tasks. It is important to note that all our results are currently in prototype stage, and before any implementation in a clinical environment, a prior impact assessment and clinical trial must be conducted. This notably includes verifying performance on other datasets that better represent patient characteristics, as well as conditioning all the hyperparameters chosen in this study on this dataset.

# 8    Acknowledgment

This work was supported by the French National Research Agency (ANR- 10-IAHU04-LIRYC). We would like to thank PNY Technology for the computing power made available to us for this study. The work of LB and GC has been partly funded by the European Union (ERC-2022-SYG-OCEAN-101071601). Views and opinions expressed are however those of the author(s) only and do not necessarily reflect those of the European Union or the European Research Council Executive Agency. Neither the European Union nor the granting authority can be held responsible for them.

---

*`https://alivecor.com/products/kardia12l`

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

# Contents

# A Theoretical appendix

## A.1 Sequential Monte Carlo samplers

### A.1.1 SMC Algorithm

In this section we first provide the SMC algorithm 1.

---

**Algorithm 1:** SMC

---

**Input:** observation $y$, number of diffusion steps $K$, number of particles $M$
*Operations involving index $i$ are repeated for $i \in [1 : M]$*
**Initialization:** $\xi_K^i \sim q_{\text{ref}}$
**for** $k = K - 1$ **to** 0 **do**
$\quad I_k^i \sim \text{Cat}\left(\{\omega_k(\xi_{k+1}^j)/\sum_{i=1}^M \omega_k(\xi_{k+1}^i)\}_{j=1}^M\right)$
$\quad \xi_k^i \sim \hat{p}_k^y(\cdot|\xi_{k+1}^{I_k^i})$
**end for**
**Output:** $\xi_0^{1:M}$

---

## A.2 MCGDiff

### A.2.1 Covariance matrix

**Simplified setting** Following [13, Section 2.1], we first give explicitly the potentials for the simplified case

$$Y = \overline{\mathrm{V}}^T X + \sigma_y \mathrm{S}^{-1}\varepsilon, \quad \varepsilon \sim \mathcal{N}(0, \mathrm{I}_{\mathsf{d_y}}), \tag{A.1}$$

where $\overline{V} \in \mathbb{R}^{\mathsf{d}_x \times \mathsf{d_y}}$ is an orthonormal matrices and $\mathrm{S} \in \mathbb{R}^{\mathsf{d_y} \times \mathsf{d_y}}$ is diagonal. For $i \in [1 : \mathsf{d_y}]$, we define $\tau_i := \min\{k \in [1 : K] | \upsilon_k > \sigma_y/s_i\}$ where $s_i$ is the $i$-th element of the diagonal matrix S. $\tau_i$ is defined such that the $i$-th coordinate of $Y$, $Y[i]$ and the $i$-th coordinate of $\overline{\mathrm{V}}^T X_{\tau_i}$ follow the same distribution. This is the fundamental idea of the MCGDiff algorithm for the noisy version and refer to [13, Section 2.1] for a detailed explanation.

We define, for $k \in [1 : K]$, $R_k \in \mathbb{R}^{\mathsf{d_y} \times \mathsf{d_y}}$ a diagonal matrix with values

$$R_k[i,i] = \begin{cases} (\upsilon_k^2 - \upsilon_{\tau_i}^2)^{1/2} & \text{if} \quad k > \tau_i, \\ \sigma_y/s_i & \text{if} \quad k \leq \tau_i. \end{cases}$$

We can finally define the MCGDiff potentials when the measurement models are of the type A.1. For $k \in [1 : K]$, define

$$g_l^y(x) := \mathcal{N}(y; \overline{\mathrm{V}}x, \mathrm{R}_k\mathrm{R}_k^T).$$

Note that if $k < \min\{\tau_i | i \in [1 : \mathsf{d_y}]\}$, then $g_k^y(x) = g_0^y(x)$.

**General** $\mathrm{A}_\theta$ **and diagonal** $\mathrm{D}_\theta$. Even though A.1 corresponds to a simplified version of 3.3, MCGDiff can also be applied to the case where $D_\theta = \sigma_y \mathrm{I}$ and thus

$$Y = \mathrm{A}_\theta X + b_\theta + \sigma_y \varepsilon$$

where $\varepsilon \sim \mathcal{N}(0_{\mathsf{d}_y}, \mathrm{I}_{\mathsf{d}_y})$ and $\sigma \geq 0$ and the singular value decomposition (SVD) $\mathrm{A}_\theta = \mathrm{U}_\theta \mathrm{S}_\theta \overline{\mathrm{V}}_\theta^T$, where $\overline{\mathrm{V}}_\theta \in \mathbb{R}^{\mathsf{d}_x \times \mathsf{d_y}}$, $\mathrm{U}_\theta \in \mathbb{R}^{\mathsf{d}_y \times \mathsf{d_y}}$ are two orthonormal matrices, and $\mathrm{S}_\theta \in \mathbb{R}^{\mathsf{d_y} \times \mathsf{d_y}}$ is diagonal.

Set $\mathsf{b} = \mathsf{d}_x - \mathsf{d_y}$. Multiplying the measurement equation by $\mathrm{S}_\theta^{-1}\mathrm{U}_\theta^T$ and substracting $\mathrm{S}_\theta^{-1}\mathrm{U}_\theta^T b_\theta$ yields

$$\mathbf{Y} := \mathrm{S}_\theta^{-1}\mathrm{U}_\theta^T (Y - b_\theta) = \overline{\mathrm{V}}_\theta X + \sigma_y \mathrm{S}_\theta^{-1}\tilde{\varepsilon}, \quad \tilde{\varepsilon} \sim \mathcal{N}(0, \mathrm{I}_{\mathsf{d_y}}).$$

Therefore, it is possible to use the potentials defined above which yields

$$g_k^{\theta,y}(x) := \mathcal{N}(y; \mathrm{A}_\theta x + b_\theta, \underbrace{\mathrm{U}_\theta \mathrm{S}_\theta^2 \mathrm{R}_{k,\theta}^2 \mathrm{U}_\theta^T}_{:=\Sigma_{k,\theta}}).$$

### A.2.2 Proposal Potential and Weight

Using conjugate formulas we compute the proposal kernel and the weights defined in Section 3 used in SMC algorithm

$$\hat{p}_{k|k+1}^{\theta,y}(x_k|x_{k+1}) = \frac{g_k^y(x_k)p_{k|k+1}(x_k|x_{k+1})}{\int g_k^y(z)p_{k|k+1}(z|x_{k+1})\mathrm{d}z}$$

$$= \mathcal{N}\left(x_k; W_{k,\theta}\left\{A_\theta^T \Sigma_{k,\theta}^{-1}(y-b_\theta) + v_k^{-2}\boldsymbol{\mu}_k(x_{k+1}, \mathcal{D}_{0|k}(x_{k+1}))\right\}, W_{k,\theta}\right),$$

$$\omega_{k+1}^{\theta,y}(x_{k+1}) = \frac{\int g_k^y(z)p_{k|k+1}(z|x_{k+1})\mathrm{d}z}{g_{k+1}^y(x_{k+1})} = \frac{\mathcal{N}(y; A_\theta\boldsymbol{\mu}_k(x_{k+1}, \mathcal{D}_{0|k}(x_{k+1})) + b_\theta, \Sigma_{k,\theta} + v_k^2 A_\theta A_\theta^T)}{\mathcal{N}(y; A_\theta x_{k+1} + b_\theta, \Sigma_{k+1,\theta})},$$

where $W_{k,\theta} := \left(v_k^{-2}\,\mathrm{I} + A_\theta^T \Sigma_{k,\theta}^{-1} A_\theta\right)^{-1} = \left(v_k^{-2}\,\mathrm{I} + \overline{V}_\theta R_{k,\theta}^{-2}\overline{V}_\theta^T\right)^{-1}$.

### A.3 Monte Carlo Expectation Maximization (MCEM)

---
**Algorithm 2:** MCEM

**Input:** observation $y$, number of diffusion steps $K$, number of particles $M$, regularization parameters $(\lambda_1, \lambda_2)$ (4.2), M step optimization parameters $(N_M, \gamma)$, total number of iterations $N_{EM}$, initial parameters $\theta_0$.

**for** $k = 1$ **to** $N_{EM}$ **do**

$\quad \xi_{1:M}^t \leftarrow \texttt{MCGDiff}(\theta_{t-1}, K, y, M)$

$\quad \theta_t \leftarrow \text{M-step}(\theta_{t-1}, \xi_{1:M}^t, (\lambda_1, \lambda_2), N_M, \gamma, y)$ (Algorithm 3)

**end for**

**Output:** $\xi_0^{1:M}$

---

---
**Algorithm 3:** M-step (implemented using [10])

**Input:** initial parameter $\theta$, particles $\xi_{1:M}$, regularization parameters $(\lambda_1, \lambda_2)$, number of gradient steps $N_M$, learning rate $\gamma$, observation $y$.

**for** $k = 1$ **to** $N_M$ **do**

$\quad \theta \leftarrow \text{Prox}_{\lambda_1\|.\|_1 + \lambda_2\|.\|_2}(\theta + \gamma\nabla F(\theta; \xi_{1:M}, y))$ (see Equation (A.2) for definition of $F$)

**end for**

**Output:** $\theta$

---

In algorithm 3, $F$ is the empirical error defined as follow for parmater $\theta$ when provided observation $y$ and particles $\xi_{1:M}$

$$F(\theta; \xi_{1:M}, y) = \frac{1}{M}\sum_{i=1}^{M}\|A_\theta\xi_i\bar{A}_\theta + B_\theta + D_\theta\epsilon\bar{D}_\theta - y\|_2^2. \tag{A.2}$$

## B   Numerical appendix

### B.1   `BeatDiff`

#### B.1.1   Preprocessing Implementation Details

Our preprocessing follows:

- Align the recording-frequency of all ECGs to 250 Hz by performing down or up sampling. Thus, two consecutive points in the ECG are separated by 4ms.

- Extract R peaks from the ECG. The first principal component is extracted channel-wise from the entire ECG. Subsequently, this extracted component is processed through a Savitzky-Golay filter, characterized by an order of 3 and a window length of 15. The extraction of R-peaks is then carried out based on the methodology proposed in [11].

- Select the window $[R-192\,\mathrm{ms}, R+512\,\mathrm{ms}]$ containing the QRS. This window corresponds to 176 time-points as $(192+512)/4 = 176$.

- ECGs are not normalized, unless otherwise specified for comparing to baseline methods or for improving visual clarity in figures.

### B.1.2 Dataset statistics

Table 6: Distribution of patients, gender and number of recorded beats among train, test and MI sets.

|                    | Train         | CV           | Test         | MI        |
|--------------------|---------------|--------------|--------------|-----------|
| All (patients)     | 22580         | 2723         | 2864         | 468       |
| Male (patients)    | 11722         | 1399         | 1497         | 343       |
| Female (patients)  | 10858         | 1324         | 1367         | 125       |
| All (beats)        | 214460        | 25694        | 27221        | 44911     |
| Mean (beats)       | 9.5 +/- 0.1   | 9.4 +/- 0.2  | 9.5 +/- 0.2  | 96 +/- 5  |

Figure 4: Female (pink), male (blue) ages histograms in training (left), test (middle), MI (right) sets.

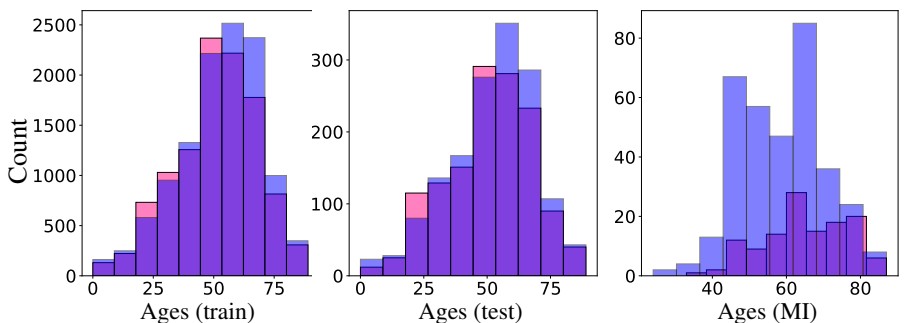

### B.1.3 Backward generation

For ECG generation in Section 4 we follow the scheduling prposed by [44] which consists of, for a total number of backward steps $\tilde{K} \in [1 : K]$, defining for $t \in [1 : \tilde{K}]$

$$v_t = \left[ v_K^{1/\rho} + \frac{t}{\tilde{K}} \left( v_1^{1/\rho} - v_K^{1/\rho} \right) \right]^{\rho}$$

Throughout our experiments, we used $\rho = 5$.

### B.1.4 Network Architecture Details

In this work, we follow closely the architecture from [24], but adapting it to the case of the ECG. Denoising diffusion consists of using a single network to learn several denoising networks, one for each level of corruption $\{v_k\}_{k=1}^{K}$ . We denote those different instances of the same network as $\{\mathcal{D}_{0|k}^{\theta}\}_{k=1}^{K}$. We describe below the different adaptations that we used for the specific case of the ECG. We start by describing how we embed the conditioning variables, namely the patient information $A, S, RR$ but also the creation of a positional embedding on the signal and the noise embedding. We then describe the parameterization used in [24] but formalized in [44] of the $F_\theta$ and finally we describe precisely the structure of the denoising network.

**Encoding of patient features** ($e_{\mathcal{P}}$)**:** As explained in Section 2, we use the $\mathcal{P} = (A, S, RR)$ as a conditioning variable for $\mathcal{D}_{0|k}$. We encode the sex S as a boolean feature $\tilde{S}$. For the numerical features, we choose the following normalization:

$$\tilde{A} = (A - 50)/50, \quad \widetilde{RR} = (RR - 500)/500 .$$

This values are chosen so that $\tilde{A} = 0$ for a 50 year old patient and $RR = 0$ if the patient heart rate is of 120 bpm. The resulting vector obtained by concatenating $\tilde{S}, \tilde{A}, \widetilde{RR}$ is fed into a two-layer dense network, yielding a $192 \times 1$ vector called $\tilde{e}_{\mathcal{P}}$. The final embedding is obtained by passing $\tilde{e}_{\mathcal{P}}$ through one MLP with SiLu activation and $2 \times 192$ neurons and a second linear layer projecting back to $\mathbb{R}^{192}$. This procedure leads to an embedding vector $e_{\mathcal{P}}$.

**Time conditioning** ($e_{\mathcal{T}}$):   We are interested in generating a fixed ECG beat. Therefore, the time wise position ($t \in [1 : \mathcal{T}]$) of each event is important to determine what is the event that we want to model in this moment. Indeed, we expect that (this can vary slightly for each signal) for $t < 50$ we observe the P-wave, for $t \in [50 : 100]$ we should observe the QRS-wave and for $t > 100$ we should observe the T-Wave. Each of this phenomenon possesses unique distinctive characteristics. This means that the data is not translation invariant. Indeed, we created each of the ECG beats by placing the R peak at the position $t = 75$.

Convolutional neural networks are translation invariant, therefore, we want to add information about the position of a certain value with respect to the whole beat window. To do so, we use positional embedding, first introduced in [91]. For a given embedding dimension $c \in \mathbb{N}$ and maximum sequence length $\mathcal{T} \in \mathbb{N}$, Positional encoding generates, for each sequence time $t \in [1 : \mathcal{T}]$ a $\mathrm{PosEnc}(t) \in \mathbb{R}^c$ by

$$\mathrm{PosEnc}(t)[l] = \begin{cases} \sin(1000^{-(2r/c)}t) & \text{if} \quad \ell = 2r\,, \\ \cos(1000^{-(2r/c)}t) & \text{if} \quad \ell = 2r + 1\,. \end{cases}$$

For the time embedding, we set $c = 192$ and we obtain a vector $e_{\mathcal{T}} = (\mathrm{PosEnc}(1), \cdots, \mathrm{PosEnc}(\mathcal{T})) \in \mathbb{R}^{\mathcal{T} \times 192}$.

**Noise level conditioning** ($e_{\upsilon_k}$):   For encoding the noise level, we follow [24] and used also Positional encoding to generate a first embedding $\tilde{e}_{\upsilon_k} = \mathrm{PosEnc}(4^{-1} \log(\upsilon_k)) \in \mathbb{R}^{192}$. The final embedding is obtained by passing $\tilde{e}_{\upsilon_k}$ through one MLP with SiLu activation and $2 \times 192$ neurons and a second linear layer projecting back to $\mathbb{R}^{192}$. This procedure leads to an embedding vector $e_{\upsilon_k}$.

**Final conditioning vector** ($e_{\mathrm{cond}}$):   We combine $(e_{\mathcal{P}}, e_{\mathcal{T}}, e_{\upsilon_k})$ into a single matrix $e_{\mathrm{cond}} \in \mathbb{R}^{\mathcal{T} \times 192}$ by broadcasting (repeating across the first dimension) $e_{\mathcal{P}}$ and $e_{\upsilon_k}$ into $(\mathcal{T}, 192)$ matrices and then defining $e_{\mathrm{cond}} := \mathrm{SiLu}(e_{\mathcal{P}} + e_{\mathcal{T}} + e_{\upsilon_k})$

**Denoising network design:**   We use the definition of the Denoising network used in [24] and which is called the F net decomposition in [44]:

$$\mathcal{D}_{0|k}^{\theta}(x, \upsilon_k, e_{\mathrm{cond}}) = c_{\mathrm{skip}}(\upsilon_k)x + c_{\mathrm{out}}(\upsilon_k)\, F_{\theta}(c_{\mathrm{in}}(\upsilon_k)x, e_{\mathrm{cond}})\,.$$

where $x$ is a $9 \times 176$ matrix corresponding to the noisy ECG beat, $c_{\mathrm{in}}(\upsilon_k) = (\upsilon_k^2 + \sigma_{\mathrm{data}}^2)^{-1/2}$, $c_{\mathrm{skip}}(\upsilon_k) = (\upsilon_k^2 + \sigma_{\mathrm{data}}^2)^{-1}\sigma_{\mathrm{data}}^2$, $c_{\mathrm{out}}(\upsilon_k) = \upsilon_k \sigma_{\mathrm{data}}(\upsilon_k^2 + \sigma_{\mathrm{data}}^2)^{-1/2}$, and $\sigma_{\mathrm{data}}$ is the (estimated) empirical standard deviation of $q_{\mathrm{data}}$. The key idea of this decomposition is that what is expected of the neural network is different for small $\upsilon_k$ and large $\upsilon_k$.

For small $\upsilon_k$, $c_{\mathrm{skip}}(\upsilon_k) \approx 1$ and $c_{\mathrm{out}}(\upsilon_k) \approx 0$, thus $\mathcal{D}_{0|k}^{\theta}(x, \upsilon_k, e_{\mathrm{cond}}) \approx x$, which is expected since $x$ is already a good reconstruction of the original data. On the contrary, when $\upsilon_k$ is large, then $c_{\mathrm{skip}}(\upsilon_k) \approx 0$ and $c_{\mathrm{out}} \approx 1$, thus $\mathcal{D}_{0|k}^{\theta}(x, \upsilon_k, e_{\mathrm{cond}})$ relies heavily on the network $F_{\theta}$ to provide a good reconstruction.

The input scaling $c_{\mathrm{in}}(\upsilon_k) = (\upsilon_k^2 + \sigma_{\mathrm{data}}^2)^{-1/2}$ is introduced so that $c_{\mathrm{in}}(\upsilon_k)x$ has always the same standard deviation. As $x$ is a realization of $X_k \sim X_0 + \upsilon_k\epsilon_k$ with $\epsilon_k \sim \mathcal{N}(0, \mathrm{I})$, $X_0 \sim q_{\mathrm{data}}$ and $\sigma_{\mathrm{data}}$ is and approximation of the standard deviation of $q_{\mathrm{data}}$, we expect $c_{\mathrm{in}}(\upsilon_k)X_k$ to have an standard deviation of approximately 1.

$F_{\theta}$ **architecture**   The $F_{\theta}$ is an evolution of the UNet firstly introduced in [71] and we follow the one proposed by [24]. We illustrate the general architecture of our $F_{\theta}$ of depth 2 in figure 5. More (or less) depth can be obtained by adding (or removing) Down blocks and Up blocks. Each block (Down, Up, Middle) is an instantiation of the general UNet block, whose architecture is shown in figure 6.

**EncoderBlock:**   The EncoderBlock consists of a 1d Convolutional layer with kernel size 3 and padding and stride 1, with 192 kernels.

**DecoderBlock:**   The Decoder block consist of a GroupNorm layer, followed by SiLu activation and finally a 1d Convolutional layer with kernel size 3 and padding and stride 1, with 9 kernels.

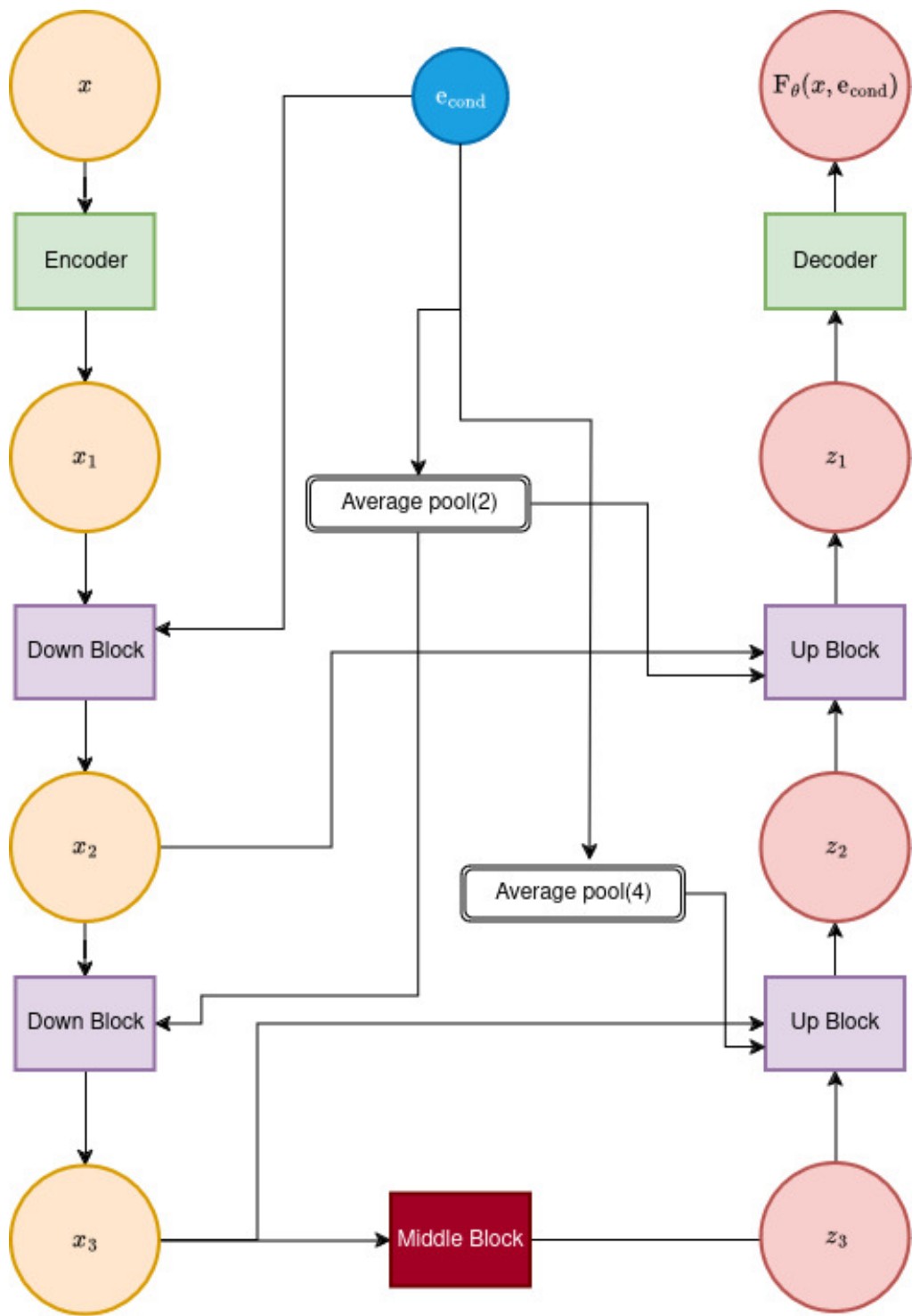

Figure 5: Illustration of $F_\theta$ architecture for a UNet of depth 2.

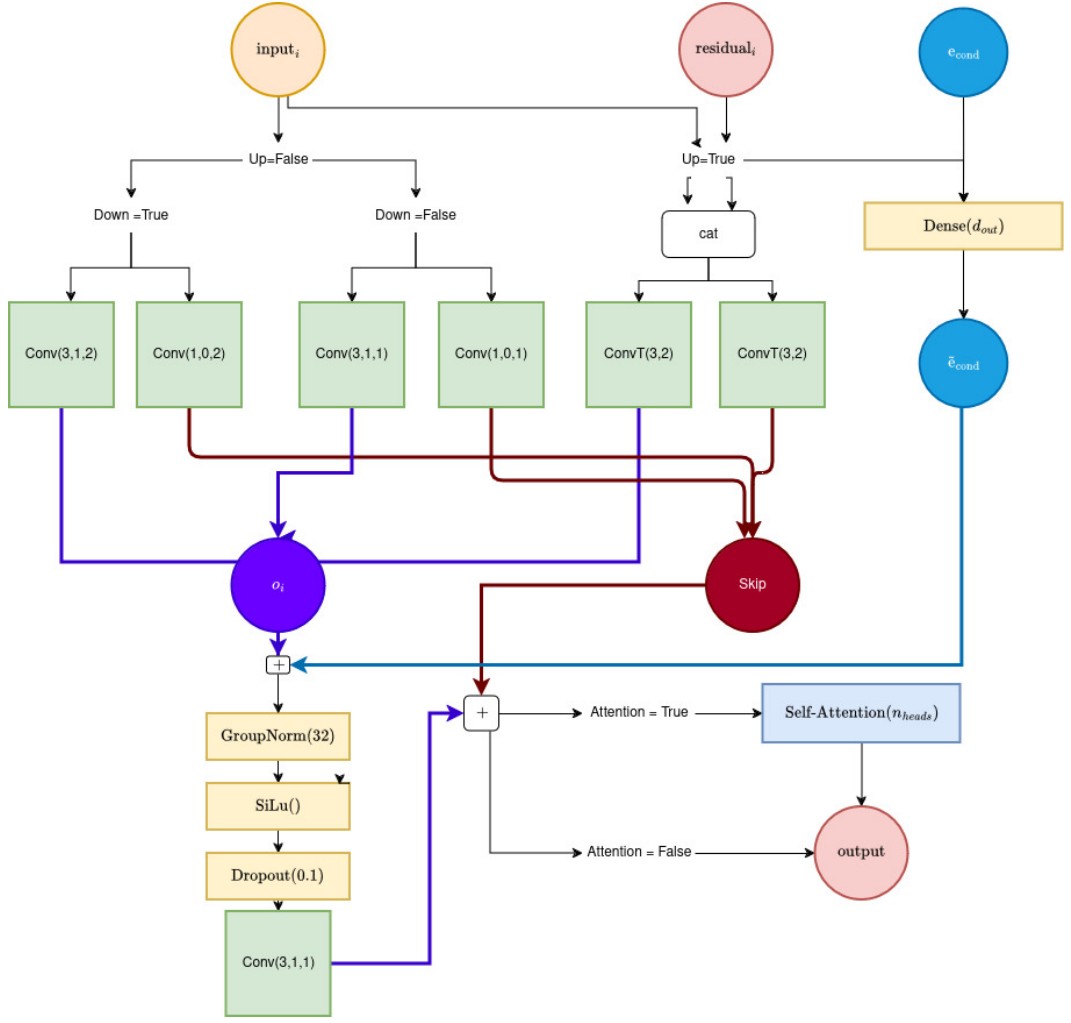

Figure 6: Illustration of a UNet block. Inputs: (Up, Down, $d_out$, Attention, $n_{heads}$).

**UNet block:** The terms in figure 6 describing the UNet with parameters (Up, Down, $d_{out}$, Attention, $n_{heads}$) block corresponds to:

- $Conv(k, p, s)$ means a 1d-convolutional layer with kernel size $k$, padding $p$ and strides $s$,
- $ConvT(k, s)$ means a 1d transposed convolutional layer with kernel size $k$ and stride $s$ using the padding configuration "Same".

The number of output channels for the convolution layers are always $d_{out}$. From UNet blocks, we can construct

- DownBlock($d_{out}, n_{heads}$, Attention): UNetBlock(False, True, $d_{out}$, Attention, $n_{heads}$),
- UpBlock($d_{out}, n_{heads}$, Attention): UNetBlock(True, False, $d_{out}$, False, $n_{heads}$),
- MiddleBlock($d_{out}, n_{heads}$, $Attention$): Stack of two UNetBlock(False, False, $d_{out}$, Attention, $n_{heads}$).

The final configuration retained for `BeatDiff` is given in Table 7. In Appendix B.1.5 we tested running a deeper architecture that is given in Table 8 . Each output from the U-Net blocks undergoes a multi-head attention layer [91], with the number of heads equal to the original dimension divided by 64. The entire network $\mathcal{D}_{0|k}^{\theta}$ is trained to minimize $\mathcal{L}_{\mathcal{D}}$ through stochastic gradient descent on the healthy training set, and the best model is selected using the cross-validation set.

| Layer Name | Parameters | Output Dimension |
|---|---|---|
| EncoderBlock | | $\mathcal{T} \times 192$ |
| DownBlock | $(d_{out}, n_{heads}, Attention) = (192, 0, False)$ | $(\mathcal{T}/2) \times 192$ |
| MiddleBlock | $(d_{out}, n_{heads}, Attention) = (192, 3, True)$ | $(\mathcal{T}/2) \times 192$ |
| UpBlock | $(d_{out}, n_{heads}, Attention) = (192, 0, False)$ | $\mathcal{T} \times 192$ |
| DecoderBlock | | $\mathcal{T} \times 9$ |

Table 7: Final configuration of `BeatDiff` .

| Layer Name | Parameters | Output Dimension |
|---|---|---|
| EncoderBlock | | $\mathcal{T} \times 192$ |
| DownBlock | $(d_{out}, n_{heads}, Attention) = (192, 0, False)$ | $(\mathcal{T}/2) \times 192$ |
| DownBlock | $(d_{out}, n_{heads}, Attention) = (384, 6, True)$ | $(\mathcal{T}/4) \times 384$ |
| MiddleBlock | $(d_{out}, n_{heads}, Attention) = (384, 6, True)$ | $(\mathcal{T}/4) \times 384$ |
| UpBlock | $(d_{out}, n_{heads}, Attention) = (192, 6, True)$ | $(\mathcal{T}/2) \times 192$ |
| UpBlock | $(d_{out}, n_{heads}, Attention) = (192, 0, False)$ | $\mathcal{T} \times 192$ |
| DecoderBlock | | $\mathcal{T} \times 9$ |

Table 8: Configuration of deeper network tested.

**Optimization:** We use the Adam optimizer [49] with the following configuration

- learning rate: $10^{-4}$,
- Number of epochs: $10^4$,
- Batch Size: 1024.

We also use exponential moving average of the network parameters with coefficient 0.9999.

**Forward diffusion parameters:** For the (forward diffusion) we used the following parameters:

- $\sigma_{\min} = 2 \times 10^{-4}$,
- $\sigma_{\max} = 80$,
- $\sigma_{\text{data}} = 0.5$,
- Importance law of $\sigma$ for training: $\mathrm{Log}\,\mathcal{N}(-1.2, 1.2^2\,\mathrm{I})$.

### B.1.5 Deeper or Unconditioned Denoisers networks

In this section we test two alternative architectures: a DDM unconditioned on the patient information $\mathcal{P}$ and a deeper DDM with configuration given in Table 8. We find that conditioning over A, S, RR leads to smaller EMD. No substantial improvements were observed when utilizing a deeper network.

### B.2 `EM-BeatDiff` parameters

### B.2.1 Number of particles

As the number of particles, denoted as $M$, increases, we observe a corresponding decrease in the discrepancy between the target posterior distribution and the distribution of particles generated by algorithm 1. A critical question arises: what is the optimal value for $M$ that strikes a balance between accuracy and computational efficiency? To approach this question, we first selected a patient from the test dataset and used algorithm 1 to generate $10^3$ samples with a high particle count of $M = 10^4$. We consider these samples as our reference representing the target posterior distribution. We then generated $10^3$ samples with algorithm 1 for different values of $M$ and calculated the Earth Mover's Distance (EMD) relative to the reference samples. This process helps us to evaluate the convergence of the distribution generated by the algorithm to the posterior as $M$ varies. Figure 7 illustrates the relationship between $M$ and the EMD. From this analysis, $M = 50$ provides an effective equilibrium

that provides a reasonable approximation to the posterior distribution while ensuring manageable inference times.

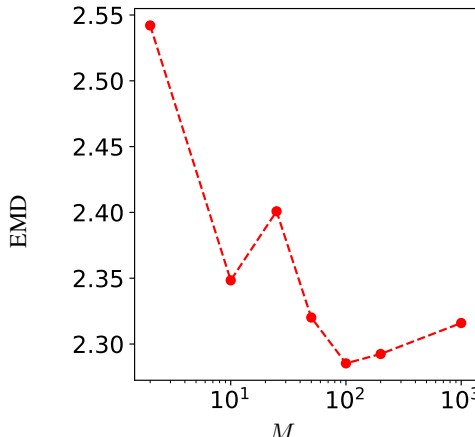

Figure 7: EMD distance between 1000 samples from algorithm 1 with $M$ particles and 1000 samples of algorithm 1 with $10^5$ particles, that is considered the standard samples.

### B.2.2 Artifacts removal parameters

**Choice of artifact basis:** We choose the following Fourier basis for removing baseline wander and electrode motion. For $j \in [1, J]$ and $t \in [1, T]$

$$c_j(t) = \begin{cases} \sin(2\frac{j}{J}t(f^a_{\max} - f^a_{\min})/f^s) & \text{if} \quad j \leq J/2\,, \\ \cos(2\frac{j}{J}t(f^a_{\max} - f^a_{\min})/f^s) & \text{else}\,, \end{cases} \tag{B.1}$$

where $J = 200$ is the number of Fourier function is the basis, $f^s = 250\text{Hz}$ is the sampling frequency, $f^a_{\min} = 0\text{Hz}$ and $f^a_{\max} = 1\text{Hz}$ is the typical range of frequency of baseline wander and electrode motion artifact.

**Regularization parameters:** In Table 9 we display the parameters used in the EM algorithm inside `EM-BeatDiff`. $(N_\text{M}, \gamma)$ indicates the number of gradient steps and the learning rate used in the M-step of the EM algorithm algorithm 2. $N_\text{EM}$ indicates the total number of EM steps used and $N_\text{M}$ the number of iterations per M step.

Table 9: Parameters used fo `EM-BeatDiff` .

| Name | $(\lambda_1, \lambda_2)$ from (4.2) | $N_\text{EM}$ | $N_\text{M}$ |
|---|---|---|---|
| QT | (0, 1) | 10 | 1 |
| AR (BW) | (10, 10) | 10 | 5 |
| AR (EM) | (10, 5) | 10 | 5 |
| ML (SW) | (0, 1) | 10 | 1 |
| ML (V1-6) | (0, 1) | 10 | 1 |
| AD (MI) | (1, 1) | 10 | 1 |
| AD (LAE) | (1, 1) | 10 | 1 |
| AD (LAD) | (1, 1) | 10 | 1 |
| AD (LQT) | (1, 1) | 10 | 1 |

### B.3 Baseline methods and networks

In this section, we provide implementation details of the adaptations that were needed to test the existing baselines to the problem in hand.

### B.3.1 WGAN [1]

In [1], the WGAN is conditioned on 15 categorical heart disease labels. These labels are embedded into a vector of size 100 and concatenated with the latent variable before being inputted into the generator. They are also embedded into a vector of length T (where T is the temporal length of the signal) and then concatenated with the cardiac signal (fake or real) before being inputted into the critic. Embedding maps variables with a finite number of possible values (i.e., categorical variables) into a vectorized representation. However, since in our DDM we condition on scalar variables($\mathcal{P} = (A, S, RR)$), in order to compare the results obtained with our DDM and the WGAN, we instead use a multi-layer perceptron (MLP) with the following architecture: a linear layer from 4 to 864, a 1D normalization layer, LeakyReLU, and a linear layer from 864 to 64. This MLP maps the 4-size feature vector $(\tilde{A}, \tilde{S}, \tilde{RR})$ to a 64-vector, which is then used in the same way as the embedding was in the original paper.

### B.3.2 SSSD [3]

We adapt the approach described in [3]. We first used the same training procedure to train a network on the same training set as ours. We added conditioning on Sex and changed the sampling frequency from 100hz to 250hz to match ours. To compare with our approach, we generate a 10s according to patient characteristics and using the NSR label from the Physionet dataset. We then use the procedure described in Appendix B.1.1 to extract heartbeats from generated ECG. For all the generation done with SSSD, we use a DDIM [83] schedule with 100 steps and $\eta = 0.01$.

### B.3.3 DeScoD [56]

The model proposed in [56] is trained to denoise the beats from the PhysioNet training set, to which noise from the MIH dataset was added. The provided code [†] was modified to train the model on 9-lead ECGs instead of 1-lead ECGs. The 9-lead preprocessed PhysioNet training set, as described in Appendix B.1.1, was used for clean ECGs, and independent random noise was added to each lead. The training procedure followed [56], where noise was sampled from 80% of the first lead of baseline wander noise from the MIT-BIH database [62] and multiplied by a random factor uniformly sampled in $[0.1, 20]$. At test time, the noise was sampled from the remaining 10% of the second lead of the noise, and no multiplication factor was used. The model was run 10 times per ECG, and the average of the 10 outputs was evaluated.

### B.3.4 EkGAN [41]

We train the model proposed in [41] to reconstruct I,II,III, V1–6 leads from I (with lr=0.0001 for 100 epochs and then applied weight decay of 0.95 per epoch). For all the inpainting experiments we normalize the ECGs by the max absolute value.

### B.3.5 AAE [76]

The model proposed in [76] was trained on the training set described in Appendix B.1.1. The architecture of the model was kept the same, except for the input channels, which were modified to $L = 9$ instead of $L = 1$.

### B.4 Classifier network for Classifier Enhancement task

The classifier used for the sex classification task is defined below, using the Flax library [36].

```python
class Classifier(nn.Module):
"""A simple CNN model."""
n_class: int = 2
@nn.compact
def __call__(self, x):
    x = nn.Conv(features=64, kernel_size=(3,))(x)
    x = nn.relu(nn.LayerNorm()(x))
    x = nn.avg_pool(x, window_shape=(2,), strides=(2,))
```

---

[†] https://github.com/HuayuLiArizona/Score-based-ECG-Denoising

```
x = nn.Conv(features=128, kernel_size=(3,))(x)
x = nn.relu(nn.LayerNorm()(x))
x = nn.avg_pool(x, window_shape=(2,), strides=(2,))
x = nn.Conv(features=256, kernel_size=(3,))(x)
x = nn.relu(nn.LayerNorm()(x))
x = nn.avg_pool(x, window_shape=(2,), strides=(2,))
x = x.mean(axis=-2)  # flatten
x = nn.Dense(features=256)(x)
x = nn.relu(nn.LayerNorm()(x))
x = nn.Dense(features=self.n_class)(x)
return x
```

All classifiers were executed with a batch size of $4096$, Adam optimizer [49] with learning rate of $0.001$ for $10^5$ steps. All classifiers achieved $100\%$ accuracy on the training set. Networks weights were initialized always using the same seed.

## B.5  Evaluation Metrics

**Sum of squared deviations (SSD):**   The sum of squared deviations between two arrays $(x, y) \in \mathbb{R}^d \times \mathbb{R}^d$ is defined as

$$\text{SSD}(x, y) = \sum_{i=1}^{d} (x_i - y_i)^2 \,.$$

**Maximum absolute deviation (MAD):**   The maximum absolute deviation between two arrays $(x, y) \in \mathbb{R}^d \times \mathbb{R}^d$ is defined as

$$\text{MAD}(x, y) = \max_{i \in [1:d]} |x_i - y_i| \,.$$

**Cosine distance (Cos.):**   The cosine distance between two arrays $(x, y) \in \mathbb{R}^d \times \mathbb{R}^d$ is defined as

$$\text{Cos.}(x, y) = \frac{x^T y}{\|x\| \|y\|} \,.$$

## B.6  Additional Results

### B.6.1  Out of distribution (OOD) score [18]

To quantify how unlikely each generated ECG is with respect to the training distribution, we used the OOD-score proposed by [18]. Their method involves using a randomly initialized network, which remains unchanged throughout the process, to produce a "random prior" by associating each training data point (images in the original paper, real or generated ECGs in our case) with a random pattern. Subsequently, a second network is trained to learn this random prior distribution, meaning that the output of the network for a training data point should be close (in terms of L2 distance) to the random pattern from the first network. After training the second network, the OOD-score for an input data point is the distance between the outputs of the two networks. The OOD-score boxplots and the resulting classification ROC curve in figure 8 show that the OOD-scores of the generated ECGs are close to those of the test ECGs, and that the scores for MI ECGs are significantly higher than those for the test and generated ECG. The authors demonstrate the relevance of their score for out-of-distribution data detection by training on four classes of the CIFAR dataset and verifying that, at test time, the score effectively distinguishes test data with the same classes as the training data from those with different classes. In our case, we adopt the same residual network architectures proposed in [18], but replace the 2D convolutions with 1D convolutions, as unidimensional residual networks are known for their efficiency in ECG classification [70]. We use 10 bootstraps and train the corresponding networks for 100 epochs with the Adam optimizer (learning rate=0.001) on healthy patients from the training set.

Table 10: $R^2$-score between QT measured vs. regressed (intercept: $\mathrm{QT}_0^c$, slope: $\mathrm{QT}_1^c$) as a function of RR, in generated samples, with 95%-CLT intervals over the test-set.

| METHOD | $R^2$-SCORE | EXPRESSION |
|---|---|---|
| Framingham | $0.88 \pm 0.03$ | $\mathrm{QT} = \mathrm{QT}_0^c + 0.154(1 - \mathrm{RR})$ |
| Bazett | $0.47 \pm 0.04$ | $\mathrm{QT} = \mathrm{QT}_1^c \sqrt{\mathrm{RR}}$ |
| Baz. (offset) | $0.98 \pm 0.00$ | $\mathrm{QT} = \mathrm{QT}_0^c + \mathrm{QT}_1^c \sqrt{\mathrm{RR}}$ |
| Fridericia | $0.94 \pm 0.02$ | $\mathrm{QT} = \mathrm{QT}_1^c \sqrt[3]{\mathrm{RR}}$ |
| Frid. (offset) | $0.98 \pm 0.00$ | $\mathrm{QT} = \mathrm{QT}_0^c + \mathrm{QT}_1^c \sqrt[3]{\mathrm{RR}}$ |

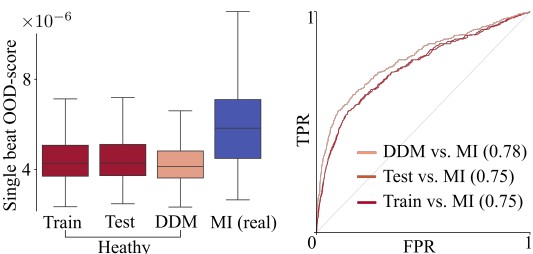

Figure 8: Out-of-distribution evaluation. **Left.** Box-plot of OOD-score for train, test, generated (Gen) and MI heart beats. **Right.** ROC curves for classification between train/test/gen and MI based on OOD-score.

### B.6.2 Prediction of QT from RR

In this section we provide supplementary results for the experiments of the prediction of corrected QT: we provide the $R^2$-score between QT measured vs. regressed (intercept: $\mathrm{QT}_0^c$, slope: $\mathrm{QT}_1^c$) as a function of RR, in generated samples, with 95%-CLT intervals over the test-set, for several corrected QT formulas in Table 10.

### B.6.3 Ablation Study cardiac anomaly

In this section, we consider for each medical condition the AUC score of the anomaly detection task while varying the way that we use the conditioning ECGs.

The configurations are shown in Table 11 and we describe now for each configuration their electrophysiological motivation.

- I, II, III: This choice of configuration implies generating the precordial leads V1–6 from the limb leads. It is coherent when the abnormality is expected to manifest in a localized way in one of the precordial leads.

- QRS: This choice of configuration implies generating the ST segment (ventricle repolarization) conditionally on the QRS observation over all the leads. It is particularly pertinent when an T-wave abnormality is expected.

- ST: This choice of configuration implies generating the QRS and P-wave from the ST segment. It is particularly coherent when the abnormality is expected in the beginning of the signal (i.e., the P-wave and QRS).

In Table 12 we display the AUC scores obtained using $1 - R^2$ between the patient signal and the mean posterior signal from EM-BeatDiff over the non-observed part of the signals.

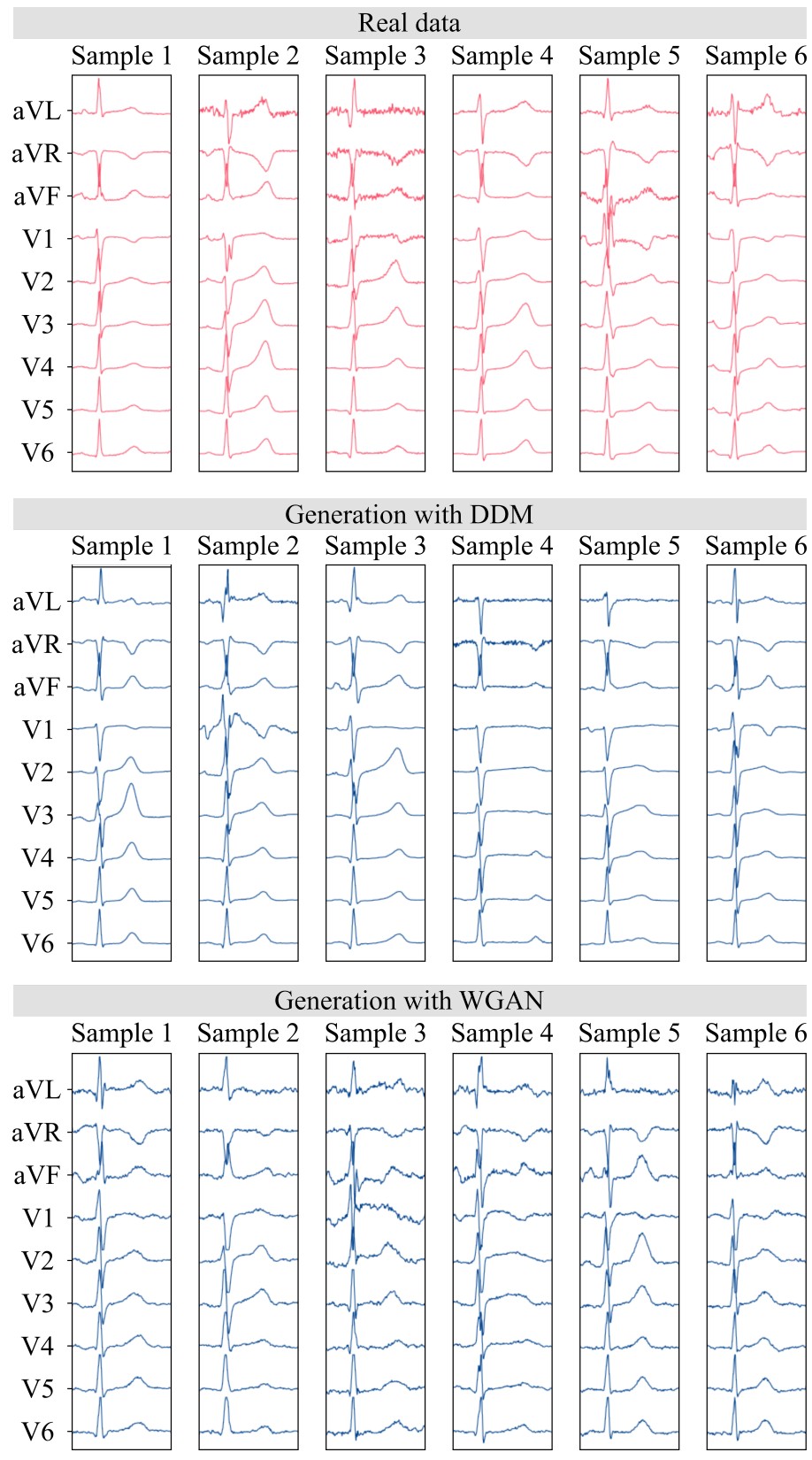

Figure 9: Real and generated ECG heart beat with DDM and WGAN.

Table 11: Configurations tested in the ablation study.

| Conditioning | $(\bar{L}, \bar{T})$ | $A_\theta$ | $\bar{A}_\theta$ | $B_\theta$ | $D_\theta$ | $\bar{D}_\theta$ |
|---|---|---|---|---|---|---|
| I, II, III | $(3, T)$ | $I_{\bar{L} \times L}$ | $I_{T \times T}$ | (4.1) | $\mathrm{diag}(\sigma_{1:\bar{L}})$ | $I_{T \times T}$ |
| QRS | $(L, 70)$ | $I_{L \times L}$ | $I_{\bar{T} \times T}$ | (4.1) | $\mathrm{diag}(\sigma_{1:L})$ | $I_{T \times \bar{T}}$ |
| ST | $(L, 106)$ | $I_{L \times L}$ | $\left[\mathbf{0}_{\bar{T}, T - \bar{T}}; I_{\bar{T} \times \bar{T}}\right]^T$ | (4.1) | $\mathrm{diag}(\sigma_{1:L}) \, I_{\bar{L} \times L}$ | $I_{\bar{T} \times T}$ |

Table 12: Anomaly detection abblation study. Confidence intervals are obtained by running 10 times `EM-BeatDiff` per heartbeat.

| Conditioning | MI | LAD | LAE | LQT |
|---|---|---|---|---|
| I, II, III | $\mathbf{84.82 \pm 0.01}$ | $91.63 \pm 0.03$ | $\mathbf{79.02 \pm 0.07}$ | $77.40 \pm 0.11$ |
| QRS | $81.88 \pm 0.02$ | $70.45 \pm 0.09$ | $62.89 \pm 0.06$ | $\mathbf{84.73 \pm 0.04}$ |
| ST | $84.05 \pm 0.01$ | $\mathbf{93.06 \pm 0.03}$ | $78.33 \pm 0.05$ | $79.72 \pm 0.06$ |

#### B.6.4 Failure cases of DeScoD in artifact removal

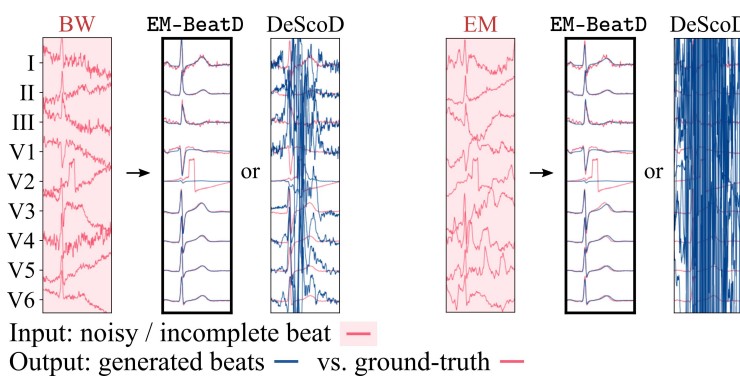

Figure 10: Failure case of DeScoD ([56]) on baseline wander (left) and electrode motion artifact (right).

Some (rare) ECGs in the test database already contain artifacts before the addition of noise from the MIT-BIH dataset. DeScoD is unable to effectively denoise these ECGs and produces inconsistent results because the noise in these ECGs is outside of the training domain of the model. The example in figure 10 illustrates that training an artifact removal model in a supervised manner is specific to the MIT-BIH database and does not allow for the removal of artifacts not found in this database, even if they share the same characteristics (low frequency). On the other hand, our approach, which is not trained in a supervised manner, is more generalizable.

#### B.6.5 10s ECGs and arrythmic data

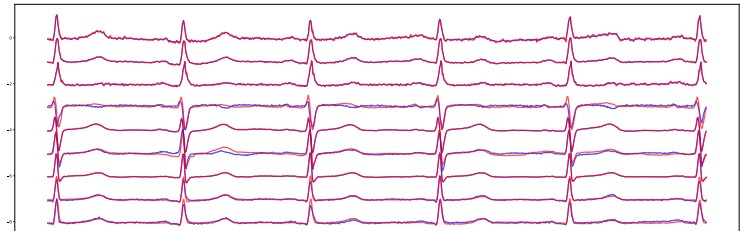

Figure 11: 9 lead (I, II, III, V1–V6 from top to bottom) 5 second healthy ECG reconstruction. Red indicates the ground-truth and blue the generated ECGs conditionned on leads I, II, III, V2 and V4.

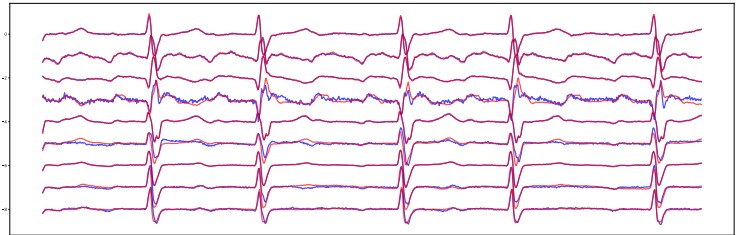

Figure 12: 9 lead (I, II, III, V1–V6 from top to bottom) 5 second AF (Atrial fibrillation) ECG reconstruction. Red indicates the ground-truth and blue the generated ECGs conditionned on leads I, II, III, V2 and V4.

## B.7 Computational resources

All the experiments were run in an internal server equipped with 8 A40 Nvidia GPUs, each with 46Gb of available memory. The server CPU has 72 threads and a total live memory of 378 Gb. All the data creation and preprocessing task were used CPU workers while all the neural network related tasks used GPU workers.

