# OpenReview forum: "Leveraging an ECG Beat Diffusion Model for Morphological Reconstruction from Indirect Signals"
_NeurIPS.cc/2024/Conference — NeurIPS 2024 poster_

### Official Review · Reviewer_TuvT · 2024-06-24

**Soundness:** 2
**Presentation:** 2
**Contribution:** 3
**Rating:** 4
**Confidence:** 4

**Summary:**

This work presents BeatDiff, a lightweight denoising diffusion generative model for multi-lead ECG signal morphology, addressing the complexities of heartbeat analysis due to noise, missing leads, and limited annotated data. It introduces the EM-BeatDiff algorithm, which leverages BeatDiff as a prior within a Bayesian framework to perform conditional generation tasks, enhancing the efficacy of ECG analysis. The combined application of BeatDiff and EM-BeatDiff demonstrates superior performance in noise removal, ECG reconstruction from single leads, and unsupervised anomaly detection compared to existing state-of-the-art methods.

**Strengths:**

This work proposes an interesting framework for ECG morphology tasks.

* Evaluation on different tasks and datasets.
* Good attempt at applying diffusion to ECG tasks.

**Weaknesses:**

Even this work explore an interesting application of diffusion model, and show multiple experimental results. here are still potential improvemnets.
* Authors reimplement EkGAN, DeScoD, SSSD, WGAN on the PhysioNet Challenge dataset, which differs from the original implementation of these baselines, making fair comparison difficult.
* There are two common baselines [1] focused on missing lead reconstruction in ECG that are not compared in this work.
* From line 128, authors claim to preprocess all ECGs from 10-second signals to single beat ECGs, and implement generation tasks on single beat ECGs rather than the original signals, reducing the novelty and applicability.
* In Table 1, the inference time of the baseline WGAN is \(3.8 \times 10^{-2}\), but the proposed work requires \(1.6 \times 10^{2}\), making it much slower than the baseline, which is inefficient and fails to outperform the baseline.


[1] Golany, Tomer, et al. "12-lead ecg reconstruction via Koopman operators." International Conference on Machine Learning. PMLR, 2021.\
[2] Chen, Jintai, et al. "ME-GAN: Learning panoptic electrocardio representations for multi-view ECG synthesis conditioned on heart diseases." International Conference on Machine Learning. PMLR, 2022.

**Questions:**

Besides the weaknesses, there are some confusing points:
* In data preprocessing, authors claim that the proposed model is trained on a healthy set. Do the authors use the annotation from the dataset to remove all unhealthy samples?
* From B1.1, authors use [R −192 ms, R + 512 ms] as the window size. Is there an ablation study exploring the effect of window size?
* Normally, ECG is 10 seconds following clinical protocol, as shown in the PTBXL and MIMIC-ECG datasets [1][2], but this work uses a single beat. Could the authors explain the reasons?


[1] Gow, B., Pollard, T., Nathanson, L. A., Johnson, A., Moody, B., Fernandes, C., Greenbaum, N., Waks, J. W., Eslami, P., Carbonati, T., Chaudhari, A., Herbst, E., Moukheiber, D., Berkowitz, S., Mark, R., & Horng, S. (2023). MIMIC-IV-ECG: Diagnostic Electrocardiogram Matched Subset (version 1.0). PhysioNet. https://doi.org/10.13026/4nqg-sb35. \
[2] Wagner, P., Strodthoff, N., Bousseljot, R., Samek, W., & Schaeffter, T. (2022). PTB-XL, a large publicly available electrocardiography dataset (version 1.0.3). PhysioNet. https://doi.org/10.13026/kfzx-aw45.

**Limitations:**

authors have already mentioned the social impact in the manuscript.

---

> ### Author Rebuttal · Authors · 2024-08-07
>
> We thank the reviewer for the feedback and the appreciation of our effort to apply and propose new methods to use diffusion models (DDM) in ECG. We now address the points raised by the reviewer.
> * *Authors reimplement EkGAN, DeScoD, SSSD, WGAN on the PhysioNet...*: We thank the reviewer their comment, we will clarify the following in the supplementary material. The pretrained weights were not available for the baselines or minor adaptation was needed to extend the networks to our case, so we had to retrain the models from scratch on the PhysionNet dataset. Our code and weights are available to ensure reproducibility and fair comparison.
> * *There are two common baselines ...*: [1] and [2] are indeed important contributions to the generative ECG community, and we regret overlooking them in our introduction. We will rectify this oversight. For the numerical comparison, we compare our work with that of Joo et al. (2023) (EkGAN in our paper), which is a more recent work that aligns with [2] as it also uses a GAN to generate 12-lead ECGs from a single lead. Unlike [2], the code for the experiments is available
> * *..to preprocess all ECGs from 10-second signals to single beat...* and *Normally, ECG is 10 seconds ... Could the authors explain the reasons?*:
>   * In this work, we focus on analyzing the morphology of heartbeats, a task that is distinct from the general ECG analysis. While the methods we present for ECG restoration and anomaly detection from partial observations can be applied, it would be necessary to define a new, presumably more complex, DDM and train it on a much larger dataset. This work is very interesting, but it differs from the work carried out in this paper.
>   * However, we want to emphasize that heartbeat analysis is an important aspect which is extremely relevant to all diseases related to the morphology of the ECG (see references [64], [65], [19], [38] in the paper). Other generative models have previously focused on a single beat, such as WGAN. Furthermore, centralizing the data is a common practice in generative models. A well known illustration is the CelebA dataset, one of the most used benchmarks in image generation tasks, where the generative models are always trained in the version of the dataset obtained by first centering all faces.
>   * We have absolutely no doubts about the contribution of the general methodology we present in the context of long ECGs [rather than heartbeats]. However, it is quite clear that a special study should be carried out to build a relevant DDM on the one hand, and to adapt the parameters of EMBeat-Diff to this new context on the other.
> Larger models allow smaller batches for parallel evaluation and thus a smaller number of maximum particles that can be used in the particle filter. However, since the distribution is more complex, we need a larger number of particles at the same time to achieve reasonable posterior sampling.
>   * We have nevertheless provided a proof of concept in those lines. We trained a DDM to produce 5-second slices instead of 10 seconds due to time and GPU constraints. We then applied "EMBeat-diff" to reconstruct 12-lead ECGs from limb leads only and I, II, III + V2 + V4 (similar to the Kardia12L mentioned by reviewer UDMo). We found that the reconstruction of the 12 leads becomes more complex when we use multi-beat ECGs, and we also need precordial leads to generate reasonable ECGs (see attached figure). We will include an improved version of this new experiment in the supplementary material to demonstrate that the inference methods we presented also apply to the analysis of 10-second segments, even though this was not the immediate aim of our study.
> * *In Table 1, the inference time of the baseline WGAN is ...*: It is well known that DDM are slower than GAN models, as they require more than one neural function evaluation (NFE). The great value of DDMs is that one can make a trade-off between quality and inference time and thereby achieve better sample quality than some state-of-the-art GAN models, as is the case in the field of computer vision [5]. In our work, we show that this is also the case for the heartbeat morphology generation of the ECG. We show that the proposed DDM outperforms WGAN in equalising a dataset (Table 1) and that it achieves a lower Earth Mover's Distance (EMD) even when performing relatively few NFEs (Figure 1).
> Furthermore, DDMs are easier to train and show better theoretical convergence results. GANs, on the other hand, are notoriously difficult to train and suffer from mode collapse.
> We would also like to point out that the WGAN has a small number of parameters. Although in principle it should be possible to obtain a better GAN network with a larger number of parameters, we consider the development of a more efficient GAN network to be outside the scope of the present work.
> * *In data preprocessing, authors claim ...*:We used the annotation from the dataset to select patients with the labels "NSR" (normal sinus rhythm), "ST" (sinus tachycardia) and "SA" (sinus arrhythmia). Thank you for pointing this out, we add a sentence in the appendix B.1.1.
> * *From B1.1, authors use [R -192 ms, R + 512 ms] ...*: Thank you for your question. We chose the window [R-192ms, R+512ms] for single beat analysis because a normal PR interval is between 120ms and 200ms (see [3]), and a normal QT interval is less than 450ms (see [4]). Anything outside this interval should not correspond to cardiac activity during a normal heartbeat.
> Additionally, the 5s ECG experiment can be considered an ablation study of the window size. It shows that reconstructing 12 leads from 5s crops is more complex and requires precordial leads for reasonable results. We will include this discussion on the choice of the window in the supplementary material.

---

> > ### Comment · Reviewer_TuvT · 2024-08-09
> >
> > Thank you for the authors' rebuttal. However, the explanations provided were cursory and unconvincing. For instance, the paper focuses on analyzing single beats rather than 10-second ECG segments. This approach relies heavily on robust external tools to segment the common 10-second clinical ECG signal into single heartbeats, which could limit the generalization of the method. Additionally, certain disease patterns, such as Premature Ventricular Contractions (PVCs), significantly alter the ECG phase, making it difficult to segment single heartbeats accurately.
> >
> > Overall, I believe that the paper is not sufficiently robust for acceptance at NeurIPS.

---

> ### Author Response · Authors · 2024-08-07
> **References to rebuttal**
>
> [1] Golany, Tomer, et al. "12-lead ecg reconstruction via Koopman operators." International Conference on Machine Learning. PMLR, 2021.
>
> [2] Chen, Jintai, et al. "ME-GAN: Learning panoptic electrocardio representations for multi-view ECG synthesis conditioned on heart diseases." International Conference on Machine Learning. PMLR, 2022.
>
> [3] Douedi S, Douedi H. P wave. [Updated 2023 Jul 24]. In: StatPearls [Internet]. Treasure Island (FL): StatPearls Publishing; 2024 Jan-. Available from: https://www.ncbi.nlm.nih.gov/books/NBK551635/
>
> [4] Goldenberg, I. L. A. N., Arthur J. Moss, and Wojciech Zareba. "QT interval: how to measure it and what is “normal”." Journal of cardiovascular electrophysiology 17.3 (2006).
>
> [5] Song, Yang, and Stefano Ermon. "Generative modeling by estimating gradients of the data distribution." Advances in neural information processing systems 32 (2019).

---

> ### Author Response · Authors · 2024-08-09
>
> We are afraid that the reviewer may have misunderstood the focus of our paper and the rebuttal.
>
> The goal of our paper is not merely the "analysis of single beats", but rather to investigate morphological heartbeat anomalies, such as Long QT syndrome (LQT) or Myocardial Infarction (MI), and to develop a white-box tool for detecting these localized anomalies. For example, LQT is only visible on the QT segment, and MI may only be visible on the precordial leads. Currently, there is no sufficiently large public dataset to train generative models for these conditions. We have developed a flexible method based on the reconstruction of 12 leads from a partial ECG, that allows us to generate the healthy counterpart of a heartbeat associated with localized MI or LQT (see Section 4). In addition to outputting an abnormality score, our approach is also able to highlight where the patient’s signal is abnormal, allowing cardiologists to rule out non-relevant abnormalities, thereby limiting the risk of hallucination.
>
> Moreover, we demonstrate that the proposed approach can be used to solve several important ECG analysis problems, such as baseline wander removal, electrode motion removal, and missing lead reconstruction. We have thoroughly evaluated the proposed technique and compared it with state-of-the-art approaches.
>
> As mentioned in the answer to UDMo’s question "how easily would it be to generate 10s ECG signals" and in the last paragraph of the rebuttal to the current reviewer TuvT, our method can be directly applied to 10-second ECGs. We have done this for healthy ECGs and ECGs with Atrial Fibrillation (see the attached PDF in the rebuttal). Furthermore, it is important to note that applying a 12 lead ECG reconstruction to arrhythmias such as Premature Ventricular Contractions (PVCs) has limited added value beyond visualization, as arrhythmia detection can be done directly by analyzing a single lead, as mentioned below.
>
> The reviewer's argument about heartbeat segmentation is unconvincing. The detection of PVC is a well-established topic, and methods for detecting PVCs have existed since 1979 (Murthy et al., Transactions on Biomedical Engineering, 1979). Today, there are methods with a sensitivity of 99.91% and specificity of 99.37% (Mazidi et al., Cluster Computing, 2020). In fact, devices like AliveCor’s KardiaMobile1L already integrate PVC detection by default. Therefore, it is straightforward to apply our heartbeat analysis to PVCs: one simply needs to detect PVCs prior to studying the heartbeats.
>
> Murthy, Ivaturi SN, and Mandayam R. Rangaraj. "New concepts for PVC detection." IEEE Transactions on Biomedical Engineering 7 (1979): 409-416.
>
> Mazidi, Mohammad Hadi, Mohammad Eshghi, and Mohammad Reza Raoufy. "Detection of premature ventricular contraction (PVC) using linear and nonlinear techniques: an experimental study." Cluster Computing 23 (2020): 759-774.

---

> > ### Comment · Reviewer_TuvT · 2024-08-11
> >
> > Thanks for the rebuttal.
> >
> > > Currently, there is no sufficiently large public dataset to train generative models for these conditions.
> > - **Question on Evaluation Data**: Does this mean all evaluation data is simulated? If true, the performance is not convincing because the performance is evaluated based on human-simulated data, but real heart behavior is too complex to be simulated by only human-designed algorithms.
> >
> > > The reviewer's argument about heartbeat segmentation is unconvincing. The detection of PVC is a well-established topic...
> > - **Reference Critique**: From the reference provided by the author, `Murthy, Ivaturi SN, and Mandayam R. Rangaraj. "New concepts for PVC detection." IEEE Transactions on Biomedical Engineering 7 (1979): 409-416.`, it mentions on Page 4, start of Section VII, "ECG recordings of two patients with PVC's were used to test the proposed scheme." Evaluating on two samples is insufficient by current research standards, as most research must be evaluated on large-scale datasets, not just a few samples.
> > - **Further Reference Analysis**: Another reference, `Mazidi, Mohammad Hadi, Mohammad Eshghi, and Mohammad Reza Raoufy. "Detection of premature ventricular contraction (PVC) using linear and nonlinear techniques: an experimental study." Cluster Computing 23 (2020): 759-774.`, mentions in Section 3.2 that only 23 samples were used to evaluate their method. This is also insufficient to demonstrate that PVC is a well-established topic, as there is no well-defined method that can detect PVC on large-scale datasets.
> > - **Benchmark Reference**: In the PTB-XL benchmark work, `Strodthoff, Nils, et al. "Deep learning for ECG analysis: Benchmarks and insights from PTB-XL." IEEE Journal of Biomedical and Health Informatics 25.5 (2020): 1519-1528.`, they provide a benchmark on multiple cardiac diseases based on ECG analysis. As shown in their work Fig 4, several diseases have AUC scores definitely lower than 95%, indicating that successfully detecting heart disease before implementing the submission method is not realistic on large-scale datasets.
> >
> > Since the author cannot address all the issues, I will keep the original score.

---

> > > ### Author Response · Authors · 2024-08-12
> > >
> > > Thank you for addressing our comment.
> > >
> > > **Question on evaluation data:**
> > >
> > > We would like to clarify that all the data used for evaluation in both the paper and the rebuttal were from real human signals and were not used during training or cross-validation.
> > >
> > > **Segmentation of single heartbeats:**
> > >
> > > The segmentation of single heartbeats is a common practice in the literature. Many works, including the baselines we analyzed in our paper, such as EkGAN and DeScoD, and more recent papers [1], rely on external tools to segment the common 10-second clinical ECG signal into single heartbeats.
> > >
> > > **Generalization to patients with PVC:**
> > >
> > > It is true that the literature on PVC detection often evaluates their works on the publicly available MIT-BIH dataset, which contains 22 records. However, it is important to note that these records last 30 minutes and correspond to approximately 3k occurrences of PVC.
> > >
> > > Additionally, a more recent study [2], evaluates their method on the Incart dataset, which contains 75 ECGs of 30 minutes, totaling approximately 20k PVCs. This further demonstrates the availability of more robust methods evaluated in substantial datasets.
> > >
> > > Moreover, in the article by Strodthoff, Nils et al. cited by the reviewer, it is indicated that "PVC is easily verifiable also for non-cardiologists." Thus, the detection of PVC prior to heartbeat segmentation is reasonable for applying the heartbeat analysis.
> > >
> > > **Benchmark Reference:**
> > >
> > > The paper by Strodthoff, Nils et al. is very interesting, and we thank the reviewer for bringing it to our attention. However, we do not fully understand how the classification of ECGs according to cardiac conditions relates to our proposed algorithm for morphological analysis of heartbeats, which does not require preliminary classification.
> > >
> > > [1] Kim, Y.; Lee, M.; Yoon, J.; Kim, Y.; Min, H.; Cho, H.; Park, J.; Shin, T. Predicting Future Incidences of Cardiac Arrhythmias Using Discrete Heartbeats from Normal Sinus Rhythm ECG Signals via Deep Learning Methods. Diagnostics 2023, 13, 2849.
> > >
> > > [2] Cai, Z.; Wang, T.; Shen, Y.; Xing, Y.; Yan, R.; Li, J.; Liu, C. Robust PVC Identification by Fusing Expert System and Deep Learning. Biosensors 2022, 12, 185.

---

### Official Review · Reviewer_sRY5 · 2024-07-11

**Soundness:** 3
**Presentation:** 4
**Contribution:** 3
**Rating:** 8
**Confidence:** 5

**Summary:**

The manuscript describes a diffusion model that provides a prior to a conditioned linear Bayesian inverse model to produce the heart beat normalized morphology of the  12 lead ECG signal from a single lead  ECG measurement.  The result is shown very promising results in finding anomalous heart, baseline wander and electron motion artifacts.

In addition the Authors bring up an extensive set of use cases (missing lead reconstruction, Anomaly detection including QT detection) for their system and compare their algorithm to state of the art solutions, GAN and  Adversarial auto encoder.

**Strengths:**

The Manuscript introduces a capability to produce the morphological features of the heart beat from single lead recordings (e.g. smart watches).

The combination of conditioned linear Bayesian inverse problem with deep learning (diffusion model) based prior distribution generation bring out good parts in both domains.

Presentation is clear and thorough going meticulously through the implementation of the diffusion model as well as the linear inversion problem. The topics delegated appendices are well selected and informative.

**Weaknesses:**

In  Table 1 BeatDiff, WGAN and SSDM are compared and it seems that WGAN has fastest inference and smallest models. SSDM seem out of league with huge model size and long latency.

Usually accuracy is improved if the parameter counts are increased, Hence one could expect that a bigger GAN - comparable to BeatDiff in memory  size and latency could achieve the performance level of BeatDiff. Why is it not so in this case?

**Questions:**

Look above.

**Limitations:**

As this is a proof of concept rather than a clinical tool, the Authors have addressed the addressed the limitations in a adequate level.

---

> ### Author Rebuttal · Authors · 2024-08-07
>
> We would like to first thank the reviewer for taking the time to review our work.
>
> Regarding the question formulated in the weakness section, indeed, it is generally accepted that increasing the number of parameters can improve model accuracy. However, in our case, we found that increasing the size of the GAN compared to the WGAN proposed in the literature made the training strategy much more complex. We did not manage to achieve better results with a larger GAN. It is possible that a different architecture and an adapted training procedure could make a larger GAN competitive with BeatDiff, but this was not the main purpose of our work. We plan to explore this avenue in subsequent work.

---

> > ### Author Response · Authors · 2024-08-12
> >
> > Dear Reviewer sRY5,
> >
> > We are grateful for your insightful review of our work. Your comments have been invaluable in helping us improve the quality and clarity of our paper. We have carefully considered each of your points and have made a thorough and comprehensive response to address your concerns (here, and in the Main Comment above).
> >
> > We would like to inquire if you have any further questions regarding our response. Your insights are valuable to us, and we greatly appreciate your attention and feedback!
> >
> > Sincerely

---

> ### Comment · Reviewer_sRY5 · 2024-08-12
> **Thank you for the clarifications.**
>
> I will retain my score.

---

### Official Review · Reviewer_h18s · 2024-07-11

**Soundness:** 2
**Presentation:** 1
**Contribution:** 2
**Rating:** 4
**Confidence:** 4

**Summary:**

Authors introduces BeatDiff based on denoising diffusion generative model and EM-BeatDiff combining BeatDiff with an Expectation-Maximization. BeatDiff is used for various ECG tasks, including noise and artifact removal, 12-lead ECG reconstruction from a single lead, and unsupervised anomaly detection. The EM-BeatDiff solves these tasks without fine-tuning, outperforming state-of-the-art methods across multiple metrics.

**Strengths:**

- Applying existing frameworks to the problem of ECG is a very good approach. Additionally, the development has been done in a highly efficient manner compared to widely used models like GANs.
- The authors have prepared a variety of tasks needed to analyze the ECG beat generation model based on a structural understanding of ECG and conducted precise analyses on these tasks.

**Weaknesses:**

- The content related to well-known methods such as DDM, Monte Carlo Guided Diffusion, and Bayesian inverse problems used in the presented model occupies too much space. It would be better to focus more specifically on the proposed methods. This also raises many questions about the experiments.
- There is a shortcoming in the selection of comparison models. Although performance evaluations were conducted on various tasks, different models were used for each task. The model should be comparable regardless of the task. A comprehensive comparison is needed. It would be beneficial to separate the diffusion-based and GAN-based models and compare their performance in detail.

**Questions:**

- There are many different ECG generation models available. What are the advantages of using a diffusion model? A performance comparison with existing models for each task is necessary. If a comparison is difficult, it would be helpful to specify the exact reasons why only the respective models were used for each task.
- The paper proposes EM-BeatDiff based on BeatDiff. Is it obvious that EM-BeatDiff performs better than using BeatDiff alone? A comparative analysis is needed.
- Conditional options such as age, gender, and RR were used. However, only the effect of RR is confirmed in the actual paper. Can other conditions not be utilized?

**Limitations:**

- The rationale for choosing diffusion models and their advantages are not clearly articulated. The paper lacks a thorough comparison with other state-of-the-art generative models, which would provide a more convincing argument for the selection of diffusion
- The dataset used in the study may be limited to specific conditions or environments. Therefore, the generalization ability of the model across diverse patient populations and different clinical settings may not be fully validated.

---

> ### Author Rebuttal · Authors · 2024-07-31
>
> We thank the reviewer for their detailed feedback.
>
> * *There is a shortcoming in the selection of comparison models. Although performance evaluations were conducted on various tasks, different models were used for each task. The model should be comparable regardless of the task. A comprehensive comparison is needed. It would be beneficial to separate the diffusion-based and GAN-based models and compare their performance in detail.*
>
> Comparison methods are specialized for specific tasks and cannot be used interchangeably. For example, the DeScoD model is designed for baseline-wander removal and cannot be used for ECG reconstruction or ECG anomaly detection, while EkGAN and AAE are not suitable for baseline-wander removal. To our knowledge, EM-BeatDiff is the only one capable of addressing all these tasks.
> For each task, we have taken care to use the most appropriate comparison methods and to compare our results with the state-of-the-art for the corresponding task.
> Could you please specify how you envision comparing all of these methods regardless of the task?
>
> * *It would be beneficial to separate the diffusion-based and GAN-based models and compare their performance in detail.*
>
> Some of the methods we have included in our study, such as the Adversarial Autoencoder (AAE), do not fall into either of these categories. Would you mind providing more details on how you envision the separation of these methods?
>
> * *There are many different ECG generation models available. What are the advantages of using a diffusion model?*
>
> Section 4 of the paper provides the advantages of using a diffusion model. In Table 1 and in Figure 1 of the paper, we provide comparisons of BeatDiff with various ECG generation models in terms of size, inference time, and different evaluation metrics. As stated on line 214, "Diffusion-based models BeatDiff and SSDM outperform WGAN, with BeatDiff being 400 times faster than SSDM".
>
> * *A performance comparison with existing models for each task is necessary. If a comparison is difficult, it would be helpful to specify the exact reasons why only the respective models were used for each task.*
>
> We already provide an exhaustive comparison of our BeatDiff model with existing ECG generation models such as SSDM or WGAN (which to the best of our knowledge are currently the state of the art). We also extensively compare our EM-BeatDiff algorithm on multiple tasks with multiple approaches that are respectively the state of the art on their tasks (such as DeScoD for baseline-wander removal, EkGAN for missing lead reconstruction, and AAE for anomaly detection). For each task, we merely selected the baseline approach which was the state-of-the-art method for this task. To better address your concerns, we are happy to provide additional comparison. Please let us know if there is a specific method that you think we should include, and on which specific task.
>
> * *The paper proposes EM-BeatDiff based on BeatDiff. Is it obvious that EM-BeatDiff performs better than using BeatDiff alone? A comparative analysis is needed.*
>
> BeatDiff and EM-BeatDiff serve different purposes and are not directly comparable. BeatDiff is a diffusion model trained to generate ECGs, while EM-BeatDiff is a sampling algorithm that utilizes BeatDiff as a prior to address various challenges in heartbeat morphology reconstruction from partial observations. As stated in the paper (line 63), "we show how BeatDiff can be used as a prior to address various challenges in heartbeat morphology reconstruction from partial observations."
> Therefore, BeatDiff alone cannot be used to solve the same tasks as EM-BeatDiff. BeatDiff serves as the generative prior for EM-BeatDiff, and the two are not meant to be compared directly. EM-BeatDiff leverages the generative capabilities of BeatDiff to perform downstream tasks such as heartbeat morphology reconstruction, which a diffusion model alone cannot accomplish.
>
> * *Conditional options such as age, gender, and RR were used. However, only the effect of RR is confirmed in the actual paper. Can other conditions not be utilized?*
>
> Age, sex, and RR all have an impact on ECG morphology. However, although age and sex have a significant impact on ECG morphology, there is no known formula in the literature that represents this correlation in a closed form. The RR interval has a more subtle impact, but it is still significant and well established in the ECG literature under the Fridericia formula (QT linearly correlated with RR^(1/3)). Furthermore, the effect of the sex variable is visible in the classifier improvement score given in Table 1. We will clarify this in the updated version of the paper. Thank you for pointing this out.
>
> * *The rationale for choosing diffusion models and their advantages are not clearly articulated. The paper lacks a thorough comparison with other state-of-the-art generative models, which would provide a more convincing argument for the selection of diffusion*
>
> c.f. response to above question on the justification of using diffusion models.
>
> * *The dataset used in the study may be limited to specific conditions or environments. Therefore, the generalization ability of the model across diverse patient populations and different clinical settings may not be fully validated.*
>
> We used the PhysioNet Challenge dataset, comprising 43k 12-lead ECGs. This dataset contains the largest and most diverse database of 12-leads ECGs publicly available. Building a larger ECG database is beyond the scope of this paper.

---

> > ### Comment · Reviewer_h18s · 2024-08-12
> >
> > For comparison models:
> >
> > 1. Generally, GANs are considered more computationally efficient compared to Diffusion Models.
> > 2. As mentioned, although some models might not directly address the paper’s tasks, a comprehensive literature review is necessary.
> >
> > Relevant Papers for Generative Models in ECG Reconstruction:
> >
> > - [1] Golany, Tomer, et al. “12-lead ECG reconstruction via Koopman operators.” International Conference on Machine Learning. PMLR, 2021.
> > - [2] Jo, Yong-Yeon, et al. “ECGT2T: Towards Synthesizing Twelve-Lead Electrocardiograms from Two Asynchronous Leads.” ICASSP.
> >
> > Dataset Consideration:
> > - [1] Gow, B., et al. (2023). MIMIC-IV-ECG: Diagnostic Electrocardiogram Matched Subset (version 1.0). PhysioNet. https://doi.org/10.13026/4nqg-sb35.
> >
> > Additional Question:
> > - There seems to be a contradiction: the proposed method is designed to detect abnormal cases, but is it unable to reconstruct 12-lead ECGs for those cases?
> >
> > I have reviewed it again, but I still intend to maintain my score.

---

> > > ### Author Response · Authors · 2024-08-12
> > >
> > > Thank you for addressing our rebuttal.
> > >
> > > As mentioned in our answer to reviewer TuVT, we regret our omission of [1], [2] and will amend this in the final version of the paper. We thank the reviewer for the dataset suggestion.
> > >
> > > Regarding your additional question, we do not see any contradiction. Our goal in the anomaly section is to investigate morphological heartbeat anomalies, such as Long QT syndrome (LQT) or Myocardial Infarction (MI), and to develop a white-box tool for detecting these anomalies. We have developed a flexible method based on the reconstruction of 12 leads from a subpart of the ECG, that allows us to generate the healthy counterpart of a heartbeat associated with localized MI or LQT (see Section 4). In addition to outputting an abnormality score, our approach is also able to highlight where the patient’s signal is abnormal, allowing cardiologists to rule out non-relevant abnormalities, thereby limiting the risk of hallucination.

---

### Official Review · Reviewer_UDMo · 2024-07-12

**Soundness:** 4
**Presentation:** 3
**Contribution:** 4
**Rating:** 7
**Confidence:** 3

**Summary:**

This paper introduces a Diffusion based generator of ECG heartbeat. The proposed technique builds on a very recently introduced sequential approach for diffusion models for linear inverse problems. The authors describe how elegantly the proposed approach can be used to solve several important ECG analysis problems (such as denoising, missing lead reconstruction, or anomaly detection)

**Strengths:**

• The paper tackles important problems for the analysis of health (ECG) data by using an interesting approach namely generation of simulated data and showcasing how it could be linked to several applications.
• The introduced method allows for an elegant solving of different problems, which makes it really appealing.
• The authors have thoroughly evaluated their proposed technique, comparing it with SOTA approach

**Weaknesses:**

• The reconstruction of missing lead is an important avenue of research. I believe that cardiologists and clinicians are doubtful it is possible to accurately reconstruct precordial leads form Einthoven leads only. In order to convince clinicians, it would have been interesting to show that it is possible to detect pathologies (such as myocardial infraction) from the reconstructed precordial leads. AliveCor recently develop a new system based on 5 leads in order to reconstruct missing leads using a generative AI approach. But in their device, they have access to at least two precordial leads, and seem to have demonstrated their ability to detect ischemia (for example) from the reduced set of leads and by using the (simulated) reconstructed leads.

**Questions:**

• Could the authors discuss how easily it would be for them it would be to generate 10s ECG signals instead of a simple heartbeat? It would be interesting to be able to generate arrhythmic data such as ECG with PVCs or AF rhythm ECG as well.
Could the authors discuss the ability to generate precordial leads for limb leads only, and how to perform a more convincing evaluation of the usability of these additional simulated leads for classification purpose.

**Limitations:**

The authors could expand a bit on potential impact of their technique, especially the risk of hallucination for generative techniques, which could lead cardiologist to misdiagnosis when visually inspecting the generated ECG signals

---

> ### Author Rebuttal · Authors · 2024-08-07
>
> *… detect pathologies (such as MI) from the reconstructed precordial leads.* In our study, we show that it's possible to generate precordial leads from limb leads for healthy patients. This is because healthy patients have correct electrical signal conduction, and our model, trained on numerous healthy ECGs, can capture this information. However, challenges arise with conduction issues like Myocardial Infarctions (MI), where part of the heart is necrotic. If the necrosis is highly localized, predicting the ECG shape in precordial leads accurately becomes difficult. To address this, the model would need to be trained on ECGs with MI. Unfortunately, there are only 500 MI patients in Physionet, which is insufficient to cover all possible MI localizations. We are currently training a generative model on 600,000 patients from a partner hospital's cardiology service to evaluate if our algorithm can generate precordial leads for patients with localized MI or Long QT. This is a future task as the data is not yet publicly available due to anonymization reasons.
>
> *AliveCor recently develop a new system based…* Many thanks for the reference to Kardia12L from AliveCor. Their AI model for 12-lead reconstruction includes two precordial leads. This is likely because they reconstruct multiple beats of higher dimension than in our case. We found that generating higher-dimensional signals with our approach requires at least two precordial leads (see below).
> Regarding MI detection from limb leads, this is an interesting question but requires a dataset of MI to develop an accurate model. To our knowledge, there is no large public dataset of ischemia (Physionet has less than 1k, insufficient for training a diffusion model), but this would be valuable for future work.
> It is worth noting that our approach has a distinct advantage over AliveCor's AI: We do not need to know which leads should be used to reconstruct the signal before training the model. Therefore, signals acquired by an AliveCor system with two precordial leads or signals acquired with smartwatches (where we can acquire I, II, III, V1, V3 and V6 depending on the placement of the smartwatch [van der Zande, Sensors 2023]) can be used.
>
> *How easily would it be to generate 10s ECG signals?* Thank you for this interesting question. Although our study focused on heartbeat morphology, we found that our approach can generate 10-second ECG signals. We trained a diffusion model to produce 5-second slices due to time and GPU constraints. We then applied our sampling algorithm to reconstruct 12-lead ECGs from limb leads only and limb leads + V2 + V4 (similar to Kardia12L). We found that reconstructing 12 leads from multi-beat ECGs is more complex and requires precordial leads for reasonable results (see attached figure). We will include an improved version of this experiment in the supplementary material to show that our methods apply to 10-second segments, though our main focus remains on heartbeat morphology and pathology detection.
>
> *It would be interesting to be able to generate arrhythmic data such as AF* Thank you for the question. While we have focused on the reconstruction of heart beats for morphologic analysis, it is important to point out that testing the ability to reconstruct arrhythmic ECGs is valuable for testing the method in a controlled environment, even if it is not necessarily of direct clinical interest. Indeed, arrhythmias like atrial fibrillation (AF) and premature ventricular contractions (PVC) can be detected from lead I alone (e.g., with an Apple Watch or Kardia), unlike morphological abnormalities such as MI or long QT, which require examining heartbeat characteristics. We trained a diffusion model on 5-second ECGs with AF from Physionet and successfully predicted a reasonable 12-lead ECG from limbs+V2+V4 (Kardia12L setting). A larger AF dataset (not yet publicly available) would yield more interesting results, but this experiment shows the method's potential for generating rhythmic abnormalities. We will include this study in the supplementary material.
>
> *discuss the ability to generate precordial leads for limb leads only…*
> * Healthy heartbeats: Our paper focuses on generating 12-lead healthy heartbeats from partial measurements (e.g., limb leads only). We show that our approach can also classify abnormal heartbeats. In Section 4, we generate healthy counterparts of abnormal heartbeats and use the distance as an anomaly score. The flexibility of our method allows the detection of various heart conditions (MI, LQT, LAD, LAE) by reconstructing 12 leads from limb leads only, QRS only, or ST only (see Table 12).
> * Unhealthy heartbeats case: Generating unhealthy heartbeats from incomplete heartbeat (e.g., limb leads) requires a large dataset of ECGs with morphological abnormalities, which is not yet publicly available. However, this would enable the use of additional simulated leads for abnormality detection from portable devices such as Smartwatch or AliveCor products.
> * 10s ECG: Generating 10s 12-lead ECGs from limb leads only, by solving an inverse problem, is non-trivial. Our current algorithm requires precordial leads (as observed by AliveCor). A special study is needed to build a relevant diffusion model and adapt the algorithm's parameters. This adaptation is complex due to the need for larger models and more particles for posterior sampling.
>
> *risk of hallucination* Our anomaly detection tool is semi-white-box: rather than outputting an abnormality score, our approach is also able to show the healthy counterparts of an abnormal signal and highlight where they differ from the patient's signals. This could theoretically enable cardiologists to rule out abnormalities that are not relevant. However, there is still a risk of hallucinations. We have shown that the generated signals are close to the real signals for healthy patients, a clinical study must be performed before clinical use. We'll mention this in the final paper.

---

> > ### Author Response · Authors · 2024-08-12
> >
> > Dear Reviewer UDMo,
> >
> > We are grateful for your insightful review of our work. Your comments have been invaluable in helping us improve the quality and clarity of our paper. We have carefully considered each of your points and have made a thorough and comprehensive response to address your concerns (here, and in the Main Comment above).
> >
> > We would like to inquire if you have any further questions regarding our response. Your insights are valuable to us, and we greatly appreciate your attention and feedback!
> >
> > Sincerely

---

### Author Rebuttal · Authors · 2024-08-07

We thank the reviewers for taking the time to give their much valued feedback to our work. We believe that the reviews and the discussion will contribute to improve the clarity of our work. We thank the reviewers for pointing out the interest, elegance and applicability of our approach. We have addressed the suggestion and questions from each reviewer individually in their dedicated rebuttal section and precised the additions that we will make in the revised version of our work.

We provide a support pdf containing figures that support the claims made in the rebuttal for the reviewers UDMo and TuvT.

---

### Decision · Program_Chairs · 2024-09-25

**Decision:**

Accept (poster)

**Comment:**

We thank the authors for their submission.  Reviewers agreed that this work was an adept application of DDMs to ECG traces with an effective extension for conditional generation.  Further, the authors provide a thorough empirical study and comparison with appropriate baseline techniques regarding the various ECG-related tasks.

Reviewers were also quite interested if this approach could generate realistic and useful samples that were more than just a single beat --- e.g., an entire 10-second recording.  Inclusion of the longer ECG experiment mentioned in the author rebuttal will make this submission more compelling.